# Partial Rejection Control for Robust Variational Inference in Sequential Latent Variable Models

## Abstract

Effective variational inference crucially depends on a flexible variational family of distributions. Recent work has explored sequential Monte-Carlo (SMC) methods to construct variational distributions, which can, in principle, approximate the target posterior arbitrarily well, which is especially appealing for models with inherent sequential structure. However, SMC, which represents the posterior using a weighted set of particles, often suffers from particle weight degeneracy, leading to a large variance of the resulting estimators. To address this issue, we present a novel approach that leverages the idea of *partial* rejection control (PRC) for developing a robust variational inference (VI) framework. In addition to developing a superior VI bound, we propose a novel marginal likelihood estimator constructed via a dice-enterprise: a generalization of the Bernoulli factory to construct unbiased estimators for SMC-PRC. The resulting variational lower bound can be optimized efficiently with respect to the variational parameters and generalizes several existing approaches in the VI literature into a single framework. We show theoretical properties of the lower bound and report experiments on various sequential models, such as the Gaussian state-space model and variational RNN, on which our approach outperforms existing methods.

## 1 Introduction

Exact inference in latent variable models is usually intractable. Markov Chain Monte-Carlo (MCMC) (Andrieu et al., 2003) and variational inference (VI) methods (Blei et al., 2017), are commonly employed in such models to make inference tractable. While MCMC has been the traditional method of choice, often with provable guarantees, optimization-based VI methods have also enjoyed considerable recent interest due to their excellent scalability on large-scale datasets. VI is based on maximizing a lower bound constructed through a marginal likelihood estimator. For latent variable models with sequential structure, sequential Monte-Carlo (SMC) (Doucet & Johansen, 2009) returns a much lower variance estimator of the log marginal likelihood than importance sampling (Bérard et al., 2014; Cérou et al., 2011). In this work, we focus our attention on designing a low variance, unbiased, and computationally efficient estimator of the marginal likelihood.

The performance of SMC based methods is strongly dependent on the choice of the proposal distribution. Inadequate proposal distributions propose values in low probability areas under the target, leading to particle depletion (Doucet & Johansen, 2009). An effective solution is to use rejection control (Liu et al., 1998; Peters et al., 2012) which is based on an *approximate* rejection sampling step within SMC to reject samples with low importance weights.

In this work, we leverage the idea of *partial* rejection control (PRC) within the framework of SMC based VI for sequential latent variable models. To this end, we construct a novel lower bound, VSMC-PRC, and propose an efficient optimization strategy for selecting the variational parameters. Compared to other recent SMC based VI approaches (Naesseth et al., 2017; Maddison et al., 2017; Le et al., 2017), our approach consists of an inbuilt accept-reject mechanism within SMC to prevent particle depletion. The use of accept-reject within SMC makes the particle weight intractable, therefore, we use a generalization of the Bernoulli factory (Asmussen et al., 1992) to construct unbiased estimators of the marginal likelihood for SMC-PRC.

Although the idea of combining VI with an inbuilt accept-reject mechanism is not new (Salimans et al., 2015; Ruiz & Titsias, 2019; Grover et al., 2018; Gummadi, 2014), a key distinction of our approach is to incorporate an accept-reject mechanism along with a resampling framework. In contrast to standard sampling algorithms that may reject the *entire* stream of particles, we use a *partial* accept-reject on the most recent update, increasing the sampling efficiency. Further, the variational framework of SMC-PRC is interesting in itself as it combines accept-reject with particle filter methods. Therefore, our proposed bound VSMC-PRC generalizes several existing approaches for example: Variational Rejection Sampling (VRS) (Grover et al., 2018), FIVO (Maddison et al., 2014), IWAE (Burda et al., 2015), and standard variational Bayes (Blei et al., 2017).

Another key distinction is that, while existing approaches using Bernoulli factory are limited to niche one-dimensional toy examples, our proposed approach is scalable. To the best of our knowledge, there is no prior work that has used Bernoulli factories for such a general case like variational recurrent neural networks (VRNN); therefore, we believe this aspect to be a significant contribution as well. The rest of the paper is organized as follows: In Section 2, we provide a brief review on SMC, partial rejection control, and dice enterprise. In Section 3, we introduce our VSMC-PRC bound and provide new theoretical insights into the Monte-Carlo estimator and design efficient ways to optimize it. Finally, we discuss related work and present experiments on the Gaussian state-space model (SSM) and VRNN.

## 2 BACKGROUND

We denote a sequence of $T$ real-valued observations as $x_{1:T} = (x_1, x_2, \ldots, x_T)$, and assume that there is an associated sequence of latent variables $z_{1:T} = (z_1, z_2, \ldots, z_T)$. We are interested in inferring the posterior distribution of the latent variables, i.e., $p(z_{1:T}|x_{1:T})$. The task is, in general, intractable. For the rest of the paper we have used some common notations from SMC and VI literature where $z_t^i$: $i^{th}$ particle at time $t$; $A_{t-1}^i$: ancestor variable for the $i^{th}$ particle at time $t$; $\theta$ and $\phi$: model and variational parameters, respectively.

### 2.1 SEQUENTIAL MONTE CARLO WITH PARTIAL REJECTION CONTROL

An SMC sampler approximates a sequence of densities $\{p_\theta(z_{1:t}|x_{1:t})\}_{t=1}^T$ through a set of $N$ weighted samples generated from a proposal distribution. Let the proposal density be

$$q_\phi(z_{1:T}|x_{1:T}) = \prod_{t=1}^T q_\phi(z_t|x_{1:t}, z_{1:t-1}). \tag{1}$$

Consider time $t-1$ at which we have uniformly weighted samples $\{N^{-1}, z_{t-1}^i, A_{t-1}^i\}_{i=1}^N$ estimating $p_\theta(z_{1:t-1}|x_{1:t-1})$. We want to estimate $p_\theta(z_{1:t}|x_{1:t})$ such that particles with a low importance weight are automatically rejected. PRC achieves this by using an *approximate* rejection sampling step (Liu et al., 1998; Peters et al., 2012). The overall procedure is as follows:

1. Generate $z_t^i \sim q_\phi(z_t|x_{1:t}, z_{1:t-1}^{A_{t-1}^i})$ where $i = 1, 2, \ldots, N$.

2. Accept $z_t^i$ with probability

$$a_{\theta,\phi}(z_t^i|z_{1:t-1}^{A_{t-1}^i}, x_{1:t}) = \left(1 + \frac{M(i, t-1)q_\phi(z_t^i|x_{1:t}, z_{1:t-1}^{A_{t-1}^i})}{p_\theta(x_t, z_t^i|x_{1:t-1}, z_{1:t-1}^{A_{t-1}^i})}\right)^{-1}, \tag{2}$$

   where $M(i, t-1)$ is a hyperparameter controlling the acceptance rate (see Proposition 3 and Section 3.3 for more details). Note that PRC applies accept-reject only on $z_t^i$, not on the entire trajectory.

3. If $z_t^i$ is rejected go to step 1.

4. The new incremental importance weight of the accepted sample is

$$\alpha_t(z_{1:t}^i) = c_t^i Z(z_{1:t-1}^{A_{t-1}^i}, x_{1:t}), \tag{3}$$

   where $c_t^i$ is

$$c_t^i = \frac{p_\theta(x_t, z_t^i|x_{1:t-1}, z_{1:t-1}^{A_{t-1}^i})}{q_\phi(z_t^i|x_{1:t}, z_{1:t-1}^{A_{t-1}^i})a_{\theta,\phi}(z_t^i|z_{1:t-1}^{A_{t-1}^i}, x_{1:t})}, \tag{4}$$

and the intractable normalization constant $Z(.)$ (For simplicity of notation, we have ignored the dependence of $Z(.)$ on $M(i, t-1)$)

$$Z(z_{1:t-1}^{A_{t-1}^i}, x_{1:t}) = \int a_{\theta,\phi}(z_t | z_{1:t-1}^{A_{t-1}^i}, x_{1:t}) q_\phi(z_t | x_{1:t}, z_{1:t-1}^{A_{t-1}^i}) dz_t. \quad (5)$$

5. Compute Monte-Carlo estimator of unnormalized weight

$$\widetilde{w}_t^i = \frac{p_\theta(x_t, z_t^i | x_{1:t-1}, z_{1:t-1}^{A_{t-1}^i}) \frac{1}{K} \sum_{k=1}^K a_{\theta,\phi}(\delta_t^{i,k} | z_{1:t-1}^{A_{t-1}^i}, x_{1:t})}{q_\phi(z_t^i | x_{1:t}, z_{1:t-1}^{A_{t-1}^i}) a_{\theta,\phi}(z_t^i | z_{1:t-1}^{A_{t-1}^i}, x_{1:t})}, \quad (6)$$

where $\delta_t^{i,k} \sim q_\phi(z_t | x_{1:t}, z_{1:t-1}^{A_{t-1}^i})$ and $k = 1, 2, \ldots, K$. Note that $\widetilde{w}_t^i$ would be essential for constructing unbiased estimator for $p_\theta(x_{1:T})$.

6. Generate ancestor variables $A_t^i$ through dice-enterprise and set new weights $w_t^i = N^{-1}$ for $i = 1, 2, \ldots, N$:

$$A_t^i \sim \text{Categorical} \left( \frac{\alpha_t(z_{1:t}^1)}{\sum_{j=1}^N \alpha_t(z_{1:t}^j)}, \frac{\alpha_t(z_{1:t}^2)}{\sum_{j=1}^N \alpha_t(z_{1:t}^j)}, \ldots, \frac{\alpha_t(z_{1:t}^N)}{\sum_{j=1}^N \alpha_t(z_{1:t}^j)} \right). \quad (7)$$

## 2.2 DICE ENTERPRISE

Simulation of ancestor variables in Eq. 7 is non-trivial due to intractable normalization constants in the incremental importance weight (see Eq. 3). Vanilla Monte-Carlo estimation of $\alpha_t(.)$ yields biased samples of ancestor variables from Eq. 7. To address this issue, we leverage a generalization of Bernoulli factory, called dice-enterprise (Morina et al., 2019). Note that multinoulli extensions of Bernoulli factory (Dughmi et al., 2017) have also been used for resampling within intractable SMC before (Schmon et al., 2019), a key distinction of our approach is to design a scalable Bernoulli factory methodology especially useful for VI applications.

Suppose we can simulate Bernoulli($p_t^i$) outcomes where $p_t^i$ is intractable. Bernoulli factory problem simulates an event with probability $f(p_t^i)$, where $f(.)$ is some desired function. In our case, the intractable coin probability $p_t^i$ is the intractable normalization constant,

$$p_t^i = Z(z_{1:t-1}^{A_{t-1}^i}, x_{1:t}) = \int a_{\theta,\phi}(z_t | z_{1:t-1}^{A_{t-1}^i}, x_{1:t}) q_\phi(z_t | x_{1:t}, z_{1:t-1}^{A_{t-1}^i}) dz_t. \quad (8)$$

Since $p_t^i \in [0, 1]$ and we can easily simulate this coin, we obtain the dice-enterprise algorithm below.

1. Required: Constants $\{c_t^i\}_{i=1}^N$ see Eq. 4.

2. Sample $C \sim \text{Categorical} \left( \frac{c_t^1}{\sum_{j=1}^N c_t^j}, \frac{c_t^2}{\sum_{j=1}^N c_t^j}, \ldots, \frac{c_t^N}{\sum_{j=1}^N c_t^j} \right)$

3. If $C = i$, generate $U_i \sim U[0, 1]$ and $z_t \sim q_\phi(z_t | x_{1:t}, z_{1:t-1}^{A_{t-1}^i})$

   - If $U_i < a_{\theta,\phi}(z_t | z_{1:t-1}^{A_{t-1}^i}, x_{1:t})$ output $i$
   - Else go to step 2

The dice-enterprise produces unbiased ancestor variables. Note that we can easily control the efficiency of the proposed dice-enterprise through the hyper-parameter $M$ (similar to Eq. 2) in contrast to existing Bernoulli factory algorithms (Schmon et al., 2019). For details on efficiency and correctness, please refer to Section 3.1 and Section 3.3.

Our proposed VSMC-PRC bound is constructed through a marginal likelihood estimator obtained by combining the SMC sampler with a PRC step and dice-enterprise. The variance of estimators obtained through SMC-PRC particle filter is usually low (Peters et al., 2012). Therefore, we expect VSMC-PRC to be a tighter bound compared to the standard SMC based bounds used in recent works (Maddison et al., 2017; Naesseth et al., 2017; Le et al., 2017). Algorithm 1 summarizes the generative process to simulate the VSMC-PRC bound. Please see Figure 1 to visualize the generative process for VSMC-PRC.

---

**Algorithm 1** Estimating the VSMC-PRC lower bound

1: **Required**: $N$, $K$, and $M$
2: **for** $t \in \{1, 2, \ldots, T\}$ **do**
3:    **for** $i \in \{1, 2, \ldots, N\}$ **do**
4:      $z_t^i, c_t^i, \widetilde{w}_t^i \sim \mathbf{PRC}\left(q, p, M(i, t-1)\right)$
5:      $z_{1:t}^i = (z_{1:t-1}^{A_{t-1}^i}, z_t^i)$
6:    **end for**
7:    **for** $i \in \{1, 2, \ldots, N\}$ **do**
8:      $A_t^i = \mathbf{DICE\text{-}ENT}\left(\{c_t^i, z_{1:t}^i\}_{i=1}^N\right)$
9:    **end for**
10: **end for**
11: **return** $\log \prod_{t=1}^T \left(\frac{1}{N} \sum_{i=1}^N \widetilde{w}_t^i\right)$
12:
13: **PRC** $\left(q, p, M(i, t-1)\right)$
14: **while** sample not accepted **do**
15:    Generate $z_t^i \sim q_\phi(z_t | x_{1:t}, z_{1:t-1}^{A_{t-1}^i})$
16:    Accept $z_t^i$ with probability
     $a_{\theta,\phi}(z_t^i | z_{1:t-1}^{A_{t-1}^i}, x_{1:t})$

17: **end while**
18: Sample $\{\delta_t^{i,k}\}_{k=1}^K \sim q_\phi(z_t | x_{1:t}, z_{1:t-1}^{A_{t-1}^i})$
19: Calculate $\widetilde{w}_t^i$ from Eq. 6
20: Calculate $c_t^i$ from Eq. 4
21: **return** $\left(z_t^i, c_t^i, \widetilde{w}_t^i\right)$
22:
23: **DICE-ENT** $\left(\{c_t^i, z_{1:t}^i\}_{i=1}^N\right)$
24: Sample $C \sim$ Multinoulli $\left(\frac{c_t^i}{\sum_{j=1}^N c_t^j}\right)_{i=1}^N$
25: **if** $C == i$ **then**
26:    Sample $U_i \sim U[0, 1]$
27:    $z_t^i \sim q_\phi(z_t | x_{1:t}, z_{1:t-1}^{A_{t-1}^i})$
28: **end if**
29: **if** $U_i < a_{\theta,\phi}(z_t^i | z_{1:t-1}^{A_{t-1}^i}, x_{1:t})$ **then**
30:    **return** $(i)$
31: **else**
32:    return **DICE-ENT** $\left(\{c_t^i, z_{1:t}^i\}_{i=1}^N\right)$
33: **end if**

---

# 3 PARTIAL REJECTION CONTROL BASED VI FOR SEQUENTIAL LATENT VARIABLE MODELS

We now show how to leverage PRC to develop a robust VI framework for sequential latent variable models. Our framework is based on the VSMC-PRC bound presented below. The complete sampling distribution of Algorithm 1 is as follows.

$$Q_{\text{VSMC-PRC}}\left(z_{1:T}^{1:N}, A_{1:T-1}^{1:N}, \delta_{1:T}^{1:N,1:K}\right) = \left(\prod_{k=1}^K \prod_{i=1}^N q_\phi(\delta_1^{i,k} | x_1) \prod_{t=2}^T \prod_{i=1}^N \prod_{k=1}^K q_\phi(\delta_t^{i,k} | x_{1:t}, z_{1:t-1}^{A_{t-1}^i})\right) \times$$

$$\left(\prod_{i=1}^N \frac{q_\phi(z_1^i | x_1) a_{\theta,\phi}(z_1^i | x_1)}{Z(x_1)} \prod_{t=1}^{T-1} \prod_{i=1}^N \text{Discrete}(A_t^i | \alpha_t) \frac{q_\phi(z_{t+1}^i | x_{1:t+1}, z_{1:t}^{A_t^i}) a_{\theta,\phi}(z_{t+1}^i | z_{1:t}^{A_t^i}, x_{1:t+1})}{Z(z_{1:t}^{A_t^i}, x_{1:t+1})}\right) \tag{9}$$

The normalization constants $Z(.)$ in Eq. 9 are intractable and have to be estimated while calculating the weights. Therefore, we introduce an extra parameter $K$, denoting the number of Monte-Carlo samples used to estimate $Z(.)$. The Monte-Carlo estimator of VSMC-PRC bound is

$$\hat{\mathcal{L}}_{\text{VSMC-PRC}}(\theta, \phi; x_{1:T}, K) = \sum_{t=1}^T \log\left(\frac{1}{N} \sum_{i=1}^N \widetilde{w}_t^i\right). \tag{10}$$

We maximize the VSMC-PRC bound with respect to model parameters $\theta$ and variational parameters $\phi$. This requires estimating the gradient the details of which are provided in Section 3.2.

## 3.1 THEORETICAL PROPERTIES

We now present properties of the Monte-Carlo estimator $\hat{\mathcal{L}}_{\text{VSMC-PRC}}$. The key variables that affect this bound are $N$ (number of samples), hyper-parameter $M$, and the number of Monte-Carlo samples used to compute the normalization constant $Z(.)$, i.e., $K$. As discussed by Bérard et al. (2014); Naesseth et al. (2017), as $N$ increases, we expect the VSMC-PRC bound to get tighter. Hence, we will focus our attention on $M$ and $K$. All the proofs can be found in the appendix.

**Proposition 1.** *Dice-enterprise produces unbiased ancestor variables. Further, let $\Lambda_t$ be the number of iterations required for generating one ancestor variable, then $\Lambda_t \sim Geom\left(\mathbb{E}[\Lambda_t]^{-1}\right)$ where*

$$\mathbb{E}[\Lambda_t] = \frac{\sum_{i=1}^N c_t^i}{\sum_{i=1}^N c_t^i Z(z_{1:t-1}^{A_{t-1}^i}, x_{1:t})}.$$

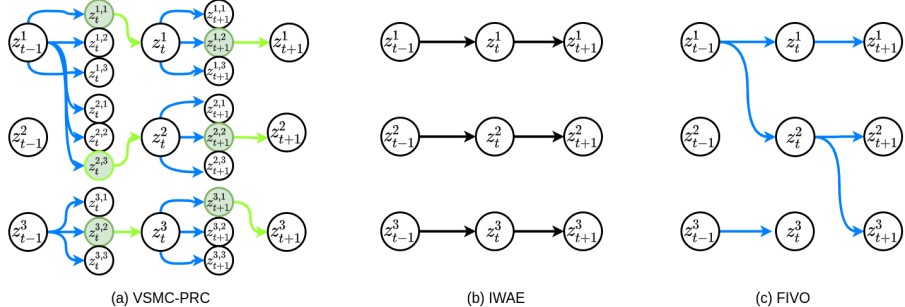

(a) VSMC-PRC        (b) IWAE        (c) FIVO

**Figure 1:** Comparison of VSMC-PRC with IWAE (Burda et al., 2015) and FIVO (Maddison et al., 2017) (a) The blue arrows represent the resampling step, We then generate multiple samples from parametrized proposal $z_t^i | z_{1:t-1}^i$ out of which one sample is accepted via PRC, depicted via green arrows. (b) In IWAE, there is no resampling step and no PRC step (c). In FIVO, there is a resampling step (blue arrows) but no PRC step.

As evident from Proposition 1, the computational efficiency of the dice-enterprise clearly relies on the normalization constant $Z(.)$. Note that the value of $Z(.)$ could be interpreted as the average acceptance rate of PRC which depends on the hyper-parameter $M(i, t-1)$. If the average acceptance rate for PRC for all particles is $\gamma$, then we can express the expected number of iterations as $\mathbb{E}[\Lambda_t^i] = \gamma^{-1}$. Therefore, the computational efficiency of dice-enterprise is similar to the PRC step and depends crucially on the hyper-parameter $M$.

**Proposition 2.** *For all $K$, $\exp(\hat{\mathcal{L}}_{VSMC\text{-}PRC})$ is unbiased, i.e., $\mathbb{E}\left[\exp(\hat{\mathcal{L}}_{VSMC\text{-}PRC})\right] = p_\theta(x_{1:T})$. Further, $\mathbb{E}[\hat{\mathcal{L}}_{VSMC\text{-}PRC}]$ is non-decreasing in $K$.*

The use of Monte-Carlo estimator in place of the true value of $Z(.)$ creates an inefficiency, as depicted by Proposition 2. The bound monotonically increases as we increase $K$ despite the use of resampling operation. It is important to note that Algorithm 1 produces an unbiased estimator of the marginal likelihood for all values of $K$.

**Proposition 3.** *Let the sampling distribution of the $i^{th}$ particle (generated via PRC) at time $t$ be $r_{\theta,\phi}(z_t | z_{1:t-1}^{A_{t-1}^i}, x_{1:t})$, then*
$$KL\left(r_{\theta,\phi}(z_t | z_{1:t-1}^{A_{t-1}^i}, x_{1:t}) \, \| \, p_\theta(z_t | z_{1:t-1}^{A_{t-1}^i}, x_{1:t})\right) \leq KL\left(q_\phi(z_t | z_{1:t-1}^{A_{t-1}^i}, x_{1:t}) \, \| \, p_\theta(z_t | z_{1:t-1}^{A_{t-1}^i}, x_{1:t})\right).$$

Proposition 3 implies that the use of the accept-reject mechanism within SMC refines the sampling distribution. Instead of accepting all samples, the PRC step ensures that only high-quality samples are accepted, leading to a tighter bound for VSMC in general (not always). We show in the appendix that when $M(i, t-1) \to \infty$, the PRC step reduces to pure rejection sampling (Robert & Casella, 2013). On the other hand, $M(i, t-1) \to 0$ implies that all samples are accepted from the proposal. Recall, $M(i, t-1)$ is a hyperparameter that can be tuned to control the acceptance rate. For more details on tuning $M$, see Section 3.3.

### 3.2 GRADIENT ESTIMATION

For tuning the variational parameters, we use stochastic optimization. Algorithm 1 produces the marginal likelihood estimator by sequentially sampling the particles, ancestor variables, and particles for the normalization constant $(z_{1:T}^{1:N}, A_{1:T-1}^{1:N}, \delta_{1:T}^{1:N,1:K})$.

When the variational distribution $q_\phi(.)$ is reparameterizable, we can make the sampling of $\delta_t^{i,k}$ independent of the model and variational parameters. However, the generated particles $z_t^i$ are not reparametrizable due to the PRC step. Finally, the ancestor variables are discrete and, therefore, cannot be reparameterized. The complete gradient can be divided into three core components (assuming $q_\phi(.)$ is reparametrizable):

$$\nabla_{\theta,\phi}\mathbb{E}[\hat{\mathcal{L}}_{\text{VSMC-PRC}}] = \mathbb{E}_{Q_{\text{VSMC-PRC}}}\left[\nabla_{\theta,\phi}\hat{\mathcal{L}}_{\text{VSMC-PRC}}(\theta,\phi; x_{1:T}, K)\right] + g_{\text{PRC}} + g_{\text{RSAMP}} \quad (11)$$

$$\approx \mathbb{E}_{Q_{\text{VSMC-PRC}}}\left[\nabla_{\theta,\phi}\hat{\mathcal{L}}_{\text{VSMC-PRC}}(\theta,\phi; x_{1:T}, K)\right]. \quad (12)$$

Note that $g_{\text{PRC}}$ and $g_{\text{RSAMP}}$ denote the score gradient of PRC and resampling step, respectively. Due to high variance, we have ignored these terms for the optimization. We have derived the full gradient and explored the gradient variance issues in the appendix. Please see Figure 2 (*left*) comparing the convergence of biased gradient vs. unbiased gradients on a toy Gaussian SSM.

## 3.3 LEARNING THE $M$ MATRIX

We use $M$ as a hyperparameter for the PRC step which controls the acceptance rate of the sampler. The basic scheme of tuning $M$ is as follows:

- Define a new random variable $F(z_{t+1}|z_{1:t}^{A_t^i}, x_{1:t+1}) = \log\left(\frac{q_\phi(z_{t+1}|x_{1:t+1}, z_{1:t}^{A_t^i})}{p_\theta(x_{t+1}, z_{t+1}|x_{1:t}, z_{1:t}^{A_t^i})}\right)$.

- Draw $z_{t+1}^j \sim q_\phi(z_{t+1}|x_{1:t+1}, z_{1:t}^{A_t^i})$ for $j = 1, 2, \ldots, J$.

- Evaluate $\gamma \in [0, 1]$ quantile value of $\{F(z_{t+1}^j|z_t^{A_t^i}, x_{1:t+1})\}_{j=1}^J$. In general for this case the acceptance rate would be around $\gamma$ for all particles.

$$\log M(i, t) = -\mathcal{Q}_{F(z_{t+1}|z_{1:t}^{A_t^i}, x_{1:t+1})}(\gamma). \tag{13}$$

- If $M$ matrix is very large then use a common $\{M(., t)\}_{t=1}^T$ for every time-step. In general, for this configuration, the acceptance rate would be *greater* than equal to $\gamma$ for all particles:

$$\log M(., t) = \min\left\{-\mathcal{Q}_{F(z_{t+1}|z_{1:t}^{A_t^i}, x_{1:t+1})}(\gamma)\right\}_{i=1}^N. \tag{14}$$

Through $\gamma$: a user parameter, we can directly control the acceptance rate. Therefore, both dice-enterprise and PRC would take around (less than) $\gamma^{-1}$ iterations to produce a sample for $M$ value learned from Eq. 13 (see Eq. 14). For implementation details please refer to the experiments.

Note that a similar scheme was also employed in Grover et al. (2018). We update $\{\{M(i, t - 1)\}_{i=1}^N\}_{t=1}^T$ dynamically once every $F$ epochs to save time. To learn more on setting hyper-parameter $M$, see Liu et al. (1998); Peters et al. (2012).

## 4 RELATED WORK AND SPECIAL CASES

There is significant recent interest in developing more expressive variational posteriors for latent variable models. There are two basic schemes for constructing tighter bounds on the log marginal likelihood: sampling-based methods (MCMC, rejection sampling) (Salimans et al., 2015; Ruiz & Titsias, 2019; Hoffman, 2017; Grover et al., 2018) or multiple samples from VI distributions to increase the flexibility (IS, SMC) (Burda et al., 2015; Maddison et al., 2017; Lawson et al., 2018; Naesseth et al., 2015). In this work, we present a unified framework for combining these two approaches, utilizing the best of both worlds. Although applying sampling-based methods on VI is useful, the density ratio between the true posterior and the improved density is often intractable. Therefore, we cannot take advantage of variance-reducing schemes like resampling, which is crucial for sequential models. We solve this issue through dice-enterprise: an extension of the Bernoulli factory.

Recently, Bernoulli factory has amassed a great interest in the area of Bayesian inference (Gonçalves et al., 2017a;b; Vats et al., 2020). Although Bernoulli factory is theoretically valuable, its applicability is severely limited due to a high rejection rate. In this paper, we have presented an approach that combines SMC with dice-enterprise for efficient implementation. A closely related work from SMC literature is Schmon et al. (2019), which also utilizies Bernoulli factories to implement unbiased resampling. However, their method is not scalable and designed particularly for partially observed diffusions. Another relevant work for unbiased estimation of the marginal likelihood is that of Kudlicka et al. (2020). In contrast to our approach, this method samples one additional particle and keeps track of the number of steps required by PRC for every time-step to obtain their unbiased estimator. The weights are tractable for Kudlicka et al. (2020) as they do not take into account the effect of the normalization constant $Z(.)$. On the other hand, we consider the effect of $Z(.)$ on the particle's weight, making resampling operation infeasible. To fix this intractability, we

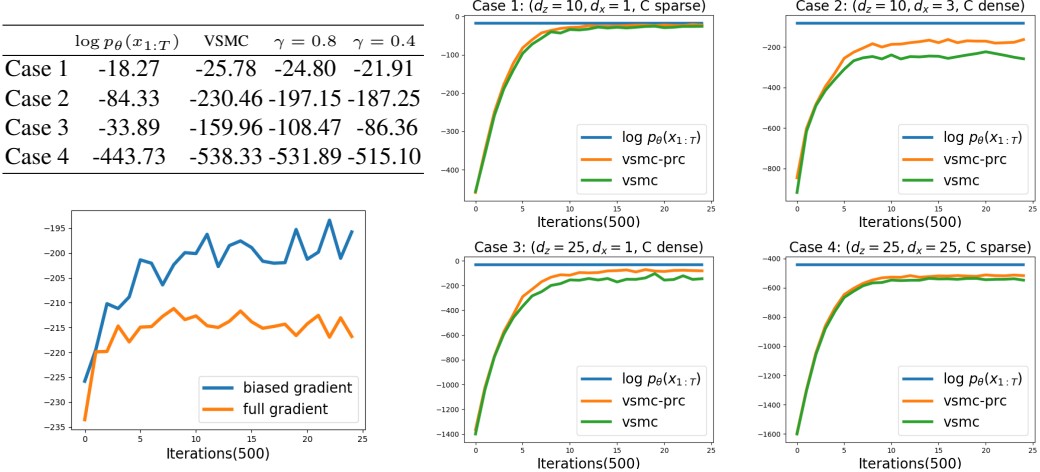

| | $\log p_\theta(x_{1:T})$ | VSMC | $\gamma = 0.8$ | $\gamma = 0.4$ |
|---|---|---|---|---|
| Case 1 | -18.27 | -25.78 | -24.80 | -21.91 |
| Case 2 | -84.33 | -230.46 | -197.15 | -187.25 |
| Case 3 | -33.89 | -159.96 | -108.47 | -86.36 |
| Case 4 | -443.73 | -538.33 | -531.89 | -515.10 |

**Figure 2:** (*Left*) The figures compares the bound value for VSMC-PRC with full gradient and biased gradient (equation 12) as a function of iterations. (*Left*) The Table compares the bound value for VSMC (Naesseth et al., 2017) and VSMC-PRC for $80\%$ and $40\%$ acceptance rate. (*Right*) We compare VSMC, VSMC-PRC ($40\%$ acceptance rate), and $\log p_\theta(x_{1:T})$ as a function of iterations.

use dice-enterprise. Comparisons of our marginal likelihood estimator versus that of Kudlicka et al. (2020) would make for interesting future work.

To provide more clarity, we will consider some special cases of VSMC-PRC bound and relate it with existing work: Note that for $N = 1$ our method reduces to a special case of Gummadi (2014) which uses a constraint function $C_t$ for every time-step and restarts the particle trajectory from $\Delta_t$ (if $C_t$ is violated). Therefore, if we use the setting $C_t(z_{1:t}) = a(z_t|z_{1:t-1}, x_{1:t})$ and $\Delta_t = t - 1$, our method reduces to a specific case of Gummadi (2014). For the special case of $N = 1$ and $T = 1$, our method reduces to VRS (Grover et al., 2018). For $N, T > 1$, if we remove the PRC step, our bound reduces to FIVO (Maddison et al., 2017). Finally, if we remove both the PRC step and resampling, then our method effectively reduces to IWAE (Burda et al., 2015). Please refer to Figure 1 for more details.

## 5 EXPERIMENTS

In this section, we evaluate our proposed algorithm on synthetic as well as real-world datasets and compare them with relevant baselines. For the synthetic data experiment, we implement our method on a *Gaussian SSM* and compare our approach with VSMC (Naesseth et al., 2017). For the real data experiment, we train a *VRNN* (Chung et al., 2015) on the polyphonic music dataset.

### 5.1 GAUSSIAN STATE SPACE MODEL

In this experiment, we study the linear Gaussian state space model. Consider the model

$$z_t = A z_{t-1} + e_z,$$
$$x_t = C z_t + e_x,$$

where $e_z, e_x \sim \mathcal{N}(0, I)$ and $z_0 = 0$. We are interested in learning a good proposal for the above model. The latent variable is denoted by $z_t$ and the observed data by $x_t$. Let the dimension of $z_t$ be $d_z$ and dimension of $x_t$ be $d_x$. The matrix $A$ has the elements $(A)_{i,j} = \alpha^{|i-j|+1}$, for $\alpha = 0.42$. We explore different settings of $d_z, d_x$, and matrix $C$. A sparse version of $C$ matrix measures the first $d_x$ components of $z_t$, on the other hand a dense version of $C$ is normally distributed i.e $C_{i,j} \sim \mathcal{N}(0, 1)$. We consider four different configurations for the experiment. For more details please refer to Figure 2.

The variational distribution is a multivariate Gaussian with unknown mean vector $\mu = \{\mu_d\}_{d=1}^{d_z}$ and diagonal covariance matrix $\{\log \sigma_d^2\}_{d=1}^{d_z}$. We set $N = 4$ and $T = 10$ for all the cases:

$$q(z_t|z_{t-1}) \sim \mathcal{N}\left(z_t|A z_{t-1} + \mu, \text{diag}(\sigma^2)\right).$$

The $\{\{M(i, t-1)\}_{i=1}^N\}_{t=1}^T$ matrix (see Eq. 13) for approximate rejection sampling is updated once every 10 epochs with acceptance rate $\gamma \in \{0.8, 0.4\}$. For estimating the intractable normalization constants, we generate $K = 3$ samples. Figure 2: (*left*) compares the convergence of biased gradient vs unbiased gradients. Note that we get a much tighter bound as compared to VSMC (Naesseth et al., 2017).

## 5.2 Variational RNN

VRNN (Chung et al., 2015) comprises of three core components: the observation $x_t$, stochastic latent state $z_t$, and a deterministic hidden state $h_t(z_{t-1}, x_{t-1}, h_{t-1})$, which is modeled through a RNN. For the experiments, we use a single-layer LSTM for modeling the hidden state. The conditional distributions $p_t(z_t|.)$ and $q_t(z_t|.)$ are assumed to be factorized Gaussians, parametrized by a single layer neural net. The output distribution $g_t(x_t|.)$ depends on the dataset. For a fair comparison, we use the same model setting as employed in FIVO (Maddison et al., 2017). We evaluate our model on four polyphonic music datasets: `Nottingham`, `JSB chorales`, `Musedata`, and `Piano-midi.de`.

Each observation $x_t$ is represented as a binary vector of 88 dimensions. Therefore, we model the observation distribution $g_t(x_t|.)$ by a set of 88 factorized Bernoulli variables. We split all four data-sets into the standard train, validation, and test sets. For tuning the learning rate, we use the validation test set. For a fair comparison, we use the same learning rate and iterations for all the models. Let the dimension of hidden state (learned by single layer LSTM) be $d_h$ and dimension of latent variable be $d_z$. We choose the setting $d_z = d_h = 64$ for all the data-sets except JSB. For modeling JSB, we use $d_z = d_h = 32$. For VSMC-PRC we have considered $N \in \{4, 6\}$ Further, for each $N$, we consider four settings $(K, \gamma) \in \{(1, 0.9), (1, 0.8), (3, 0.9), (3, 0.8)\}$. The $M$ hyper-parameter for PRC step is learned from Eq. 14 due to large size. We have updated $M$ value once every 50 epoches. Note that in this scenario, the acceptance rate for all particles would be *greater* than equal to $\gamma$. For more details on experiments, please refer to the appendix.

As discussed in Section 3.1, the PRC step and dice-enterprise have time complexity $\mathcal{O}(N/\gamma)$ for producing $N$ samples (assuming average acceptance rate $\gamma$). Therefore, we consider $\lceil N\gamma^{-1} \rceil$ particles for IWAE and FIVO to ensure effectively the same number of particles, where $N \in \{4, 6\}$ and $\gamma = 0.8$. Note, however, that the acceptance rate is $\geq \gamma$, so this adjustment actually favors the other approaches more. For FIVO, we perform resampling when ESS falls below $N/2$. Table 1 summarizes the results which show whether rejecting samples provide us with any benefit or not, and as the results show, our approach, even with the aforementioned adjustment, outperforms the other approaches in terms of test log-likelihoods, while still having a similar computational cost.

**Table 1:** We report Test log-likelihood for models trained with FIVO, IWAE, ELBO, and VSMC-PRC. For VSMC-PRC $N = (4, 6)$ and $(K, \gamma) \in \{(1, 0.9), (1, 0.8), (3, 0.9), (3, 0.8)\}$ (results are in this order). The results for pianoroll data-sets are in nats per timestep.

| N | Data | ELBO | IWAE | FIVO | N | VSMC-PRC | | | |
|---|---|---|---|---|---|---|---|---|---|
| | Nott | -3.87 | -3.12 | -3.07 | | **-2.96** | -2.98 | -2.99 | -2.96 |
| | jsb | -8.69 | -8.01 | -7.51 | | -7.41 | **-7.28** | -7.37 | -7.36 |
| 5 | Piano | -7.99 | -7.97 | -7.85 | 4 | -7.82 | -7.86 | **-7.80** | -7.85 |
| | Muse | -7.48 | -7.45 | -6.75 | | -6.61 | -6.63 | -6.66 | **-6.58** |
| N | Data | ELBO | IWAE | FIVO | N | VSMC-PRC | | | |
| | Nott | -3.87 | -3.87 | -2.99 | | -2.93 | -2.93 | **-2.90** | -2.91 |
| | jsb | -8.69 | -8.32 | -7.40 | | -7.29 | -7.21 | -7.16 | **-7.14** |
| 8 | Piano | -7.99 | -8.04 | -7.80 | 6 | -7.78 | -7.77 | -7.79 | **-7.77** |
| | Muse | -7.48 | -7.41 | -6.67 | | -6.60 | **-6.57** | -6.61 | -6.60 |
| Avg. Rank | | $6.87 \pm 0.33$ | $6.12 \pm 0.33$ | $4.87 \pm 0.33$ | | $2.87 \pm 1.05$ | $2.62 \pm 1.21$ | $2.87 \pm 1.26$ | $\mathbf{1.75 \pm 0.66}$ |

In Sec. 3.1, we discussed the effect of $K$ and PRC rejection rate on VSMC-PRC bound. We expect a performance improvement when $K$ and the rejection rate is increased. Although the results for VSMC-PRC's different configurations are almost the same, we still get the best average ranking for $(K = 3, \gamma = 0.8)$. Overall, for most cases, VSMC-PRC bound performs better than FIVO (Maddison et al., 2017) and IWAE (Burda et al., 2015) for a variety of configurations.

In VSMC-PRC, improvement in the bound value comes at the cost of estimating the normalization constant $Z(.)$, i.e., $K$. On further inspection, we can clearly see that increasing $K$ doesn't provide us

with any substantial benefits despite the increase in computational cost. Therefore, to maintain the computational trade-off ($K = 1, \gamma > 0.8$) seems to be a reasonable choice for VI practitioners.

Table 1 signifies that rejecting samples with low importance weight is better instead of keeping a large number of particles (at least for a reasonably high acceptance rate $\gamma$). The proposed bound uses more particles (PRC step and dice-enterprise) than existing approaches like FIVO and IWAE due to intractability. Future work aims at designing a scalable implementation for VSMC-PRC bound that consumes fewer particles.

## 6 CONCLUSION

We introduced VSMC-PRC, a novel bound that combines SMC and *partial* rejection sampling with VI in a synergistic manner. This results in a robust VI procedure for sequential latent variable models. Instead of using standard sampling algorithms, we have employed a partial sampling scheme suitable for high dimensional sequences. Our experimental results clearly demonstrate that VSMC-PRC outperforms existing bounds like IWAE (Burda et al., 2015) and standard particle filter bounds (Maddison et al., 2017; Naesseth et al., 2017; Le et al., 2017). The future work aims to explore *partial* versions of powerful sampling algorithms like Hamiltonian Monte Carlo (Neal et al., 2011) instead of rejection sampling.

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

# A    PROOF OF THEORETICAL RESULTS

*Proof of proposition 1* : Dice-enterprise produces unbiased ancestor variables. Let's evaluate the probability that Dice-enterprise would output $i$ as the ancestor index. Assume that the algorithm terminates after $r$ steps where $r \in \{1, 2, \ldots, \infty\}$, then the probability of output $i$ is given as follows:

$$
\begin{aligned}
\Pr(\text{output} = i) &= \sum_{r=1}^{\infty} \Pr(\text{output} = i | \text{after } r \text{ steps}) \\
&= \frac{c_t^i Z(z_{1:t-1}^{A_{t-1}^i}, x_{1:t})}{\sum_{m=1}^{N} c_t^m} \sum_{r=1}^{\infty} \left( \sum_{j=1}^{N} \frac{c_t^j (1 - Z(z_{1:t-1}^{A_{t-1}^j}, x_{1:t}))}{\sum_{m=1}^{N} c_t^m} \right)^{r-1} \\
&= \frac{c_t^i Z(z_{1:t-1}^{A_{t-1}^i}, x_{1:t})}{\sum_{j=1}^{N} c_t^j Z(z_{1:t-1}^{A_{t-1}^j}, x_{1:t})} .
\end{aligned}
$$

It is easy to see that $\Lambda_t$ is geometrically distributed with success probability given by

$$
\Pr(\text{getting output in a loop}) = \frac{\sum_{i=1}^{N} c_t^i Z(z_{1:t-1}^{A_{t-1}^i}, x_{1:t})}{\sum_{m=1}^{N} c_t^m}
$$

*Proof of proposition 2* : Before explaining the proof, we will first introduce $\widehat{Z}$: Monte-Carlo estimator of the unknown normalization constant $Z(.)$. Since we are using $K$ samples

$$
\widehat{Z}(x_{1:t}, z_{1:t-1}^{A_{t-1}^i}; K) = \frac{1}{K} \sum_{k=1}^{K} a_{\theta,\phi}(\delta_t^{i,k} | x_{1:t}, z_{1:t-1}^{A_{t-1}^i}). \tag{15}
$$

Algorithm 1 is producing an unbiased estimator of the marginal likelihood. We will first integrate out $\delta_{1:T}^{1:N,1:K}$ from the marginal likelihood estimator.

$$
\begin{aligned}
&\mathbb{E}_{Q_{\text{VSMC-PRC}}} \left[ \exp(\hat{\mathcal{L}}_{\text{VSMC-PRC}}) \right] \\
&= \sum_{A_{1:T-1}^{1:N}} \int \prod_{t=1}^{T} \frac{1}{N} \sum_{i=1}^{N} \frac{p_\theta(x_t, z_t^i | x_{1:t-1}, z_{1:t-1}^{A_{t-1}^i}) \widehat{Z}(x_{1:t}, z_{1:t-1}^{A_{t-1}^i}; K)}{q_\phi(z_t^i | x_{1:t}, z_{1:t-1}^{A_{t-1}^i}) a_{\theta,\phi}(z_t^i | x_{1:t}, z_{1:t-1}^{A_{t-1}^i})} \\
&\qquad Q_{\text{VSMC-PRC}} \left( z_{1:T}^{1:N}, A_{1:T-1}^{1:N}, \delta_{1:T}^{1:N,1:K} \right) dz_{1:T}^{1:N} d\delta_{1:T}^{1:N,1:K} \\
&= \sum_{A_{1:T-1}^{1:N}} \int \prod_{t=1}^{T} \frac{1}{N} \sum_{i=1}^{N} \frac{p_\theta(x_t, z_t^i | x_{1:t-1}, z_{1:t-1}^{A_{t-1}^i}) \int \widehat{Z}(x_{1:t}, z_{1:t-1}^{A_{t-1}^i}; K) \prod_{k=1}^{K} q_\phi(\delta_t^{i,k} | x_{1:t}, z_{1:t-1}^{A_{t-1}^i}) d\delta_t^{i,k}}{q_\phi(z_t^i | x_{1:t}, z_{1:t-1}^{A_{t-1}^i}) a_{\theta,\phi}(z_t^i | x_{1:t}, z_{1:t-1}^{A_{t-1}^i})} \\
&\qquad Q_{\text{VSMC-PRC}} \left( z_{1:T}^{1:N}, A_{1:T-1}^{1:N} \right) dz_{1:T}^{1:N} \\
&= \sum_{A_{1:T-1}^{1:N}} \int \prod_{t=1}^{T} \frac{1}{N} \sum_{i=1}^{N} \frac{p_\theta(x_t, z_t^i | x_{1:t-1}, z_{1:t-1}^{A_{t-1}^i}) Z(x_{1:t}, z_{1:t-1}^{A_{t-1}^i})}{q_\phi(z_t^i | x_{1:t}, z_{1:t-1}^{A_{t-1}^i}) a_{\theta,\phi}(z_t^i | x_{1:t}, z_{1:t-1}^{A_{t-1}^i})} Q_{\text{VSMC-PRC}} \left( z_{1:T}^{1:N}, A_{1:T-1}^{1:N} \right) dz_{1:T}^{1:N} \\
&= \sum_{A_{1:T-1}^{1:N}} \int \prod_{t=1}^{T} \frac{1}{N} \sum_{i=1}^{N} \frac{p_\theta(x_t, z_t^i | x_{1:t-1}, z_{1:t-1}^{A_{t-1}^i})}{r_{\theta,\phi}(z_t^i | x_{1:t}, z_{1:t-1}^{A_{t-1}^i})} Q_{\text{VSMC-PRC}} \left( z_{1:T}^{1:N}, A_{1:T-1}^{1:N} \right) dz_{1:T}^{1:N} \\
&= \sum_{A_{1:T-1}^{1:N}} \int \prod_{t=1}^{T} \frac{1}{N} \sum_{i=1}^{N} w_t^i \left( \prod_{i=1}^{N} r_{\theta,\phi}(z_1^i | x_1) \prod_{t=1}^{T-1} \prod_{i=1}^{N} \text{Discrete}(A_t^i | \alpha_t) r_{\theta,\phi}(z_{t+1}^i | x_{1:t+1}, z_{1:t}^{A_t^i}) \right) dz_{1:T}^{1:N} \\
&= \mathbb{E} \left[ \prod_{t=1}^{T} \frac{1}{N} \sum_{i=1}^{N} w_t^i \right] . \tag{16}
\end{aligned}
$$

Note that $r_{\theta,\phi}(z_t^i|.)$ is the sampling density of PRC. It is easy to see that Eq. 16 is a standard SMC estimator for the marginal likelihood $p_\theta(x_{1:T})$. The proof is quite standard in SMC literature and can be found in (Naesseth et al., 2017; 2019). The key factor that makes our bound unbiased is the ability to produce unbiased ancestor samples despite the presence of intractable normalization constant. Using Jensen inequality we can easily show that VSMC-PRC bound is smaller then log marginal likelihood.

$$\mathbb{E}\left[\hat{\mathcal{L}}_{\text{VSMC-PRC}}; K\right] = \int \sum_{t=1}^{T} \log \left[\frac{1}{N} \sum_{i=1}^{N} \frac{p_\theta(x_t, z_t^i|x_{1:t-1}, z_{1:t-1}^{A_{t-1}^i})\widehat{Z}(x_{1:t}, z_{1:t-1}^{A_{t-1}^i}; K)}{q_\phi(z_t^i|x_{1:t}, z_{1:t-1}^{A_{t-1}^i})a_{\theta,\phi}(z_t^i|x_{1:t}, z_{1:t-1}^{A_{t-1}^i})}\right]$$
$$Q_{\text{VSMC-PRC}}\left(z_{1:T}^{1:N}, A_{1:T-1}^{1:N}, \delta_{1:T}^{1:N,1:K}\right) dz_{1:T}^{1:N} d\delta_{1:T}^{1:N,1:K} dA_{1:T-1}^{1:N} \leq \log p_\theta(x_{1:T}). \tag{17}$$

We will show that $\mathbb{E}[\hat{\mathcal{L}}_{\text{VSMC-PRC}}; K]$ is non-decreasing with $K$. Let's define a collection of subsets $\{\{I_{i,t}\}_{i=1}^N\}_{t=1}^T \subset \{1, 2, \ldots, K\}$ having elements $\{i_1, i_2, \ldots, i_m\}$ randomly drawn from the set $\{1, 2, \ldots, K\}$ having length $m \leq K$. We can easily show the following expectation:

$$\widehat{Z}(x_{1:t}, z_{1:t-1}^{A_{t-1}^i}; K) = \mathbb{E}_{I_{i,t}=\{i_1, i_2, \ldots, i_m\}}\left[\widehat{Z}(x_{1:t}, z_{1:t-1}^{A_{t-1}^i}; m)\right] = \frac{1}{K}\sum_{k=1}^{K} a_{\theta,\phi}(\delta_t^{i,k}|x_{1:t}, z_{1:t-1}^{A_{t-1}^i}) \tag{18}$$

Substitute the values of above expectation in equation 17. Use Jensen's inequality i.e $\mathbb{E}[\log X] \leq \log \mathbb{E}[X]$ to complete the proof .

$$\mathbb{E}\left[\hat{\mathcal{L}}_{\text{VSMC-PRC}}; K\right]$$

$$= \mathbb{E}\left[\sum_{t=1}^{T} \log \mathbb{E}_{\{\{I_{i,t}\}_{i=1}^N\}_{t=1}^T}\left(\frac{1}{N}\sum_{i=1}^{N}\frac{p_\theta(x_t, z_t^i|x_{1:t-1}, z_{1:t-1}^{A_{t-1}^i})\widehat{Z}(x_{1:t}, z_{1:t-1}^{A_{t-1}^i}; m)}{q_\phi(z_t^i|x_{1:t}, z_{1:t-1}^{A_{t-1}^i})a_{\theta,\phi}(z_t^i|x_{1:t}, z_{1:t-1}^{A_{t-1}^i})}\right)\right]$$

$$\geq \mathbb{E}\left[\mathbb{E}_{\{\{I_{i,t}\}_{i=1}^N\}_{t=1}^T}\sum_{t=1}^{T} \log\left(\frac{1}{N}\sum_{i=1}^{N}\frac{p_\theta(x_t, z_t^i|x_{1:t-1}, z_{1:t-1}^{A_{t-1}^i})\widehat{Z}(x_{1:t}, z_{1:t-1}^{A_{t-1}^i}; m)}{q_\phi(z_t^i|x_{1:t}, z_{1:t-1}^{A_{t-1}^i})a_{\theta,\phi}(z_t^i|x_{1:t}, z_{1:t-1}^{A_{t-1}^i})}\right)\right]$$

$$\geq \mathbb{E}\left[\hat{\mathcal{L}}_{\text{VSMC-PRC}}; m\right]$$

Now we will see what happens when the limit $K \to \infty$. Using *dominated convergence theorem* we can write down the estimator as follows:

$$\lim_{K\to+\infty}\mathbb{E}[\hat{\mathcal{L}}_{\text{VSMC-PRC}}]$$

$$= \lim_{K\to+\infty}\mathbb{E}\left[\sum_{t=1}^{T}\log\left(\frac{1}{N}\sum_{i=1}^{N}\frac{p_\theta(x_t, z_t^i|x_{1:t-1}, z_{1:t-1}^{A_{t-1}^i})\widehat{Z}(x_{1:t}, z_{1:t-1}^{A_{t-1}^i}; K)}{q_\phi(z_t^i|x_{1:t}, z_{1:t-1}^{A_{t-1}^i})a_{\theta,\phi}(z_t^i|x_{1:t}, z_{1:t-1}^{A_{t-1}^i})}\right)\right]$$

$$= \mathbb{E}\left[\sum_{t=1}^{T}\log\left(\frac{1}{N}\sum_{i=1}^{N}\frac{p_\theta(x_t, z_t^i|x_{1:t-1}, z_{1:t-1}^{A_{t-1}^i})\lim_{K\to+\infty}\widehat{Z}(x_{1:t}, z_{1:t-1}^{A_{t-1}^i}; K)}{q_\phi(z_t^i|x_{1:t}, z_{1:t-1}^{A_{t-1}^i})a_{\theta,\phi}(z_t^i|x_{1:t}, z_{1:t-1}^{A_{t-1}^i})}\right)\right]$$

$$= \mathbb{E}\left[\sum_{t=1}^{T}\log\left(\frac{1}{N}\sum_{i=1}^{N}\frac{p_\theta(x_t, z_t^i|x_{1:t-1}, z_{1:t-1}^{A_{t-1}^i})Z(x_{1:t}, z_{1:t-1}^{A_{t-1}^i})}{q_\phi(z_t^i|x_{1:t}, z_{1:t-1}^{A_{t-1}^i})a_{\theta,\phi}(z_t^i|x_{1:t}, z_{1:t-1}^{A_{t-1}^i})}\right)\right] \leq \log p_\theta(x_{1:T})$$

*Proof of proposition 3* : We will show that PRC step refines the learned distribution.

$$KL\left(r(z_t|z_{1:t-1}^{A_{t-1}^i}, x_{1:t})||p(z_t|z_{1:t-1}^{A_{t-1}^i}, x_{1:t})\right)$$

$$= \int \log\left(\frac{r(z_t|z_{1:t-1}^{A_{t-1}^i}, x_{1:t})}{p(z_t|z_{1:t-1}^{A_{t-1}^i}, x_{1:t})}\right) r(z_t|z_{1:t-1}^{A_{t-1}^i}, x_{1:t}) dz_t$$

$$= \int \log\left(\frac{q(z_t|z_{1:t-1}^{A_{t-1}^i}, x_{1:t}) a(z_t|z_{1:t-1}^{A_{t-1}^i}, x_{1:t})}{p(z_t|z_{1:t-1}^{A_{t-1}^i}, x_{1:t}) Z(x_{1:t}, z_{1:t-1}^{A_{t-1}^i})}\right) r(z_t|z_{1:t-1}^{A_{t-1}^i}, x_{1:t}) dz_t$$

$$\leq \int \log\left(\frac{q(z_t|z_{1:t-1}^{A_{t-1}^i}, x_{1:t}) a(z_t|z_{1:t-1}^{A_{t-1}^i}, x_{1:t})}{p(z_t|z_{1:t-1}^{A_{t-1}^i}, x_{1:t}) Z(x_{1:t}, z_{1:t-1}^{A_{t-1}^i})}\right) q(z_t|z_{1:t-1}^{A_{t-1}^i}, x_{1:t}) dz_t$$

$$\leq KL\left(q(z_t|z_{1:t-1}^{A_{t-1}^i}, x_{1:t})||p(z_t|z_{1:t-1}^{A_{t-1}^i}, x_{1:t})\right) +$$

$$\int \log\left(\frac{a(z_t|z_{1:t-1}^{A_{t-1}^i}, x_{1:t})}{Z(x_{1:t}, z_{1:t-1}^{A_{t-1}^i})}\right) q(z_t|z_{1:t-1}^{A_{t-1}^i}, x_{1:t}) dz_t$$

$$\leq KL\left(q(z_t|z_{1:t-1}^{A_{t-1}^i}, x_{1:t})||p(z_t|z_{1:t-1}^{A_{t-1}^i}, x_{1:t})\right)$$

First, we will use the property of negatively correlated random variables. Note that the following two random variables

$$X = \log\left(\frac{r(z_t|z_{1:t-1}^{A_{t-1}^i}, x_{1:t})}{p(z_t|z_{1:t-1}^{A_{t-1}^i}, x_{1:t})}\right) \text{ and } Y = a(z_t|z_{1:t-1}^{A_{t-1}^i}, x_{1:t}),$$

are negatively correlated. We know that for negatively correlated variables following identity holds $\mathbb{E}[XY] \leq \mathbb{E}[X]\mathbb{E}[Y]$. Further we have used Jensen's inequality to show that

$$\mathbb{E}[\log a(z_t|z_{1:t-1}^{A_{t-1}^i}, x_{1:t})] \leq \log Z(x_{1:t}, z_{1:t-1}^{A_{t-1}^i})$$

**Case 1**: $M(i, t-1) \to 0$ implies $r(z_t|z_{1:t-1}^{A_{t-1}^i}, x_{1:t}) \to q(z_t|z_{1:t-1}^{A_{t-1}^i}, x_{1:t})$

In this situation all samples would be accepted. Hence, we can express the sampling distribution as:

$$r(z_t|z_{1:t-1}^{A_{t-1}^i}, x_{1:t}) = q(z_t^i|x_{1:t}, z_{1:t-1}^{A_{t-1}^i}), \tag{19}$$

**Case 2**: $M(i, t-1) \to \infty$ implies $r(z_t|z_{1:t-1}^{A_{t-1}^i}, x_{1:t}) \to p(z_t|z_{1:t-1}^{A_{t-1}^i}, x_{1:t})$

In this case, the acceptance probability would reduce to standard rejection sampling. Therefore, the sampling distribution would become equal to the true posterior.

$$r(z_t|z_{1:t-1}^{A_{t-1}^i}, x_{1:t}) = \frac{q(z_t^i|x_{1:t}, z_{1:t-1}^{A_{t-1}^i}) \frac{p(z_t^i, x_t|x_{1:t-1}, z_{1:t-1}^{A_{t-1}^i})}{M(i,t-1)q(z_t^i|x_{1:t}, z_{1:t-1}^{A_{t-1}^i})}}{\int q(z_t^i|x_{1:t}, z_{1:t-1}^{A_{t-1}^i}) \frac{p(z_t^i, x_t|x_{1:t-1}, z_{1:t-1}^{A_{t-1}^i})}{M(i,t-1)q(z_t^i|x_{1:t}, z_{1:t-1}^{A_{t-1}^i})} dz_t^i}$$

$$= p(z_t^i|x_{1:t}, z_{1:t-1}^{A_{t-1}^i})$$

## B  GRADIENT ESTIMATION

In this section, we will derive the unbiased gradients for the Monte-Carlo estimate $\mathbb{E}[\hat{\mathcal{L}}_{\text{VSMC-PRC}}]$. Note that we can express the complete gradient into three core components (assuming $q()$ is reparametriz-

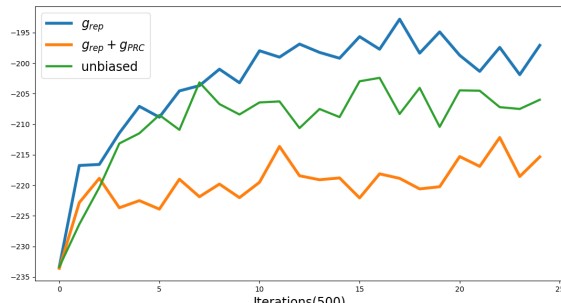

**Figure 3:** Convergence rate for biased gradient vs. unbiased gradient on toy Gaussian SSM. We compared $g_{\text{rep}}$ (blue), $g_{\text{rep}} + g_{\text{PRC}}$ (orange), and unbiased gradient (green) for optimization.

able):

$$
\nabla_{\theta,\phi}\mathbb{E}[\hat{\mathcal{L}}_{\text{VSMC-PRC}}] \;=\; g_{\text{rep}} + g_{\text{PRC}} + g_{\text{RSAMP}}
$$

$$
g_{\text{rep}} \;=\; \mathbb{E}_{Q_{\text{VSMC-PRC}}}\left[\nabla_{\theta,\phi}\hat{\mathcal{L}}_{\text{VSMC-PRC}}(\theta,\phi;x_{1:T},K)\right],
$$

$$
g_{\text{PRC}} \;=\; \mathbb{E}_{Q_{\text{VSMC-PRC}}}\left[\hat{\mathcal{L}}_{\text{VSMC-PRC}}(\theta,\phi;x_{1:T},K)\nabla_{\theta,\phi}\left(\sum_{i=1}^{N}\sum_{t=1}^{T}\log\frac{a(z_t^i|x_{1:t},z_{1:t-1}^{A_{t-1}^i})}{Z(x_{1:t},z_{1:t-1}^{A_{t-1}^i})}\right)\right],
$$

$$
g_{\text{RSAMP}} \;=\; \mathbb{E}_{Q_{\text{VSMC-PRC}}}\left[\hat{\mathcal{L}}_{\text{VSMC-PRC}}(\theta,\phi;x_{1:T},K)\nabla_{\theta,\phi}\sum_{i=1}^{N}\sum_{t=1}^{T-1}A_t^i\log\alpha_t\right]
$$

Note that for both $g_{\text{PRC}}$ and $g_{\text{RSAMP}}$ unbiased score gradient estimates are **not available** due to intractability. Therefore, we have used Monte-Carlo samples to estimate the log density for figure 3.

## C   EXPERIMENTAL SETUP

For the real data experiment, we train a *VRNN* (Chung et al., 2015) on the polyphonic music dataset. Polyphonic music comprises of four datasets: : Nottingham, JSB chorales, Musedata, and Piano-midi.de. Each dataset was divided into standard train, validation, and test datset.

The validation data was used to tune the learning rate: we picked the learning rate from the following set: $\{3\times10^{-4}, 1\times10^{-4}, 3\times10^{-5}, 1\times10^{-5}\}$, instead of optimizing for each method, we picked the learning rate at which FIVO (Maddison et al., 2017) validation performance is the best. Once the learning rate is decided, we ran every method for the same number of iterations to ensure uniformity. We use a single-layer LSTM for modeling the hidden state having dimension $d_h$. For a length $T$ sequence, the variational distribution and joint data likelihood are defined as follows

$$
r(z_{1:T}|x_{1:T}) = \frac{\prod_{t=1}^{T}q_t(z_t|h_t(z_{t-1},x_{t-1},h_{t-1}),x_t)a_t(z_t|h_t(z_{t-1},x_{t-1},h_{t-1}),x_t)}{\prod_{t=1}^{T}Z_t(h_t(z_{t-1},x_{t-1},h_{t-1}),x_t)}
$$

Note that $a_t(z_t|.)$ is the acceptance probability for the PRC step and $Z_t(.)$ is the intractable normalization constant. Similarly, we can write down the joint data likelihood as

$$
p(z_{1:T},x_{1:T}) = \prod_{t=1}^{T}p_t(z_t|h_t(z_{t-1},x_{t-1},h_{t-1}),x_t)g_t(x_t|h_t(z_{t-1},x_{t-1},h_{t-1}),z_t)
$$

The conditional distributions $p_t(z_t|.)$ and $q_t(z_t|.)$ are assumed to be factorized Gaussians, where dimension of the latent variable $z_t$ is $d_z$. Note that the conditional densities are parametrized by fully connected neural networks with a single layer having size $d_h$. The output distribution $g_t(x_t|.)$ is modeled by a set of 88 iid Bernoulli variables. Please see Table 2 for more details regarding implementation details. The unknown weights and biases are initialized using a Xavier initialization. For setting up the optimization we used a batch size of 4 with adam optimizer. The unknown hyperparameter $M$ is updated once every 50 epoches through Eq. 14 to save time.

**Table 2:** Training parameters used for VSMC-PRC, FIVO, and IWAE.

| Parameters | Datasets | | | |
|---|---|---|---|---|
| | **Piano-mid** | **Muse** | **jsb** | **Nott** |
| Optimizer | Adam | Adam | Adam | Adam |
| Learning Rate | 0.00003 | 0.0001 | 0.00003 | 0.0001 |
| Batch-size | 4 | 4 | 4 | 4 |
| $d_h$ | 64 | 64 | 32 | 64 |
| $d_z$ | 64 | 64 | 32 | 64 |
| num-iteration | 50,000 | 1,00,000 | 1,00,000 | 1,00,000 |

