# OpenReview forum: "Partial Rejection Control for Robust Variational Inference in Sequential Latent Variable Models"
_ICLR.cc/2021/Conference — Reject_

### Official Review · AnonReviewer4 · 2020-10-23
**A fine paper on Sequential Monte Carlo with partial rejection control, but the question might have been tackled in the literature already**

**Rating:** 5
**Confidence:** 4

**Review:**

The paper considers SMC to construct variational approximations. SMC methods
can be improved with partial rejection control (PRC). Then it is not obvious
that one can obtain unbiased estimators of the normalizing constant, as in
plain SMC. The authors consider a way of obtaining unbiased estimators in that
setting.  Experiments include a linear Gaussian state-space model and recurrent
neural networks on polyphonic music datasets.

---

The problem is interesting. The notation is a bit cumbersome at times, but it is
the case for most papers on SMC. The writing is mostly clear. The experiments
include a mix of toy and more realistic examples.

The fact that SMC with PRC can still produce unbiased estimators of the
marginal likelihood was described by Kudlicka, Murray, Schoen, Lindsten,
"Particle filter with rejection control and unbiased estimator of the marginal
likelihood". The authors do cite that paper, but I did not understand exactly
why that paper does not completely solve the problem that the authors consider.
At first glance it seems that the Kudlicka et al paper would apply "out of the
box" when replacing the prior by an arbitrary proposal in SMC, i.e. using a
proposal "$q_t$" instead of "$f_t$" and the ratio "$g_t f_t/q_t$" for the weights.
My understanding is that Kudlicka et al focus on the bootstrap particle filter
because PRC is a remarkably generic approach to improve it, applicable even
when sampling from the model prior is the only option; not because the same
reasoning would not apply for generic proposals.

Thus it was not clear to me that there's a need for another paper showing that
SMC with PRC can still provide unbiased estimators of the marginal likelihood.
If the authors made a convincing case that the extension to general proposals
within SMC with PRC requires significant work, their contribution would be more
convincing.

Furthermore, the manuscript makes references to Bernoulli factories and dice
enterprise.  In fact, the manuscript addresses the problem of categorical
sampling with unbiased estimators of the underlying probabilities.  The problem
was addressed in "Bernoulli Race Particle Filters" by Schmon, Doucet,
Deligiannidis.  That paper is cited by the authors, but the authors do not make
it clear that their algorithm (in Section 2.2) is exactly the same as
"Algorithm 2" of that paper, their Proposition 1 is exactly "equation (12)" of
that paper, etc.  Furthermore, I don't think their algorithm is indeed a "dice
enterprise" as in the terminology of Morina et al; I believe the references to
the Bernoulli factory literature would be sending most readers in the wrong
direction.

Based on these flaws I do not think that the manuscript is suitable for
publication. Perhaps a clearer explanation of the specific shortcomings of
Kudlicka et al's work would make the paper more convincing.

---
Small comments:

- page 2: "We further assume that the joint density [...]"
in fact that decomposition always holds, it is not an assumption.
It is just p(a,b) = p(a)p(b|a).

- page 2: A SMC sampler -> An SMC sampler

- page 2 and later: there is some inconsistency between the notation M(i,t-1) and M(t-1,i).

- The latex command \eqref seems to have been used instead of \ref, in various places.

- page 6 "utlizing the best of both worlds"

---

> ### Author Response · Authors · 2020-11-17
> **Response to AnonReviewer4**
>
> Thank you for the helpful comments. Our response to your comments is provided below:
>
> \textbf{Concerns regarding novelty}:
> Firstly, note that the goal of our paper was to propose a family of bounds that could improve Variational inference (VI), whereas Kudlicka et al (2020) does not consider VI. We agree with the reviewer that the work of Kudlicka et al. is also viable for SMC-PRC, just like our method. However, our work aims to leverage the robust nature of SMC-PRC within a variational inference framework (resulting in our framework VSMC-PRC) whereas Kudlicka et al (2020) do not consider VI in the first place but only focus on SMC-PRC. Therefore, the scope and positioning of our work is closer to recent work on VI based bounds like FIVO, IWAE, unlike SMC based methods like Kudlicka et al. We have clarified this fact further in the revised manuscript as well.
>
>  To elaborate further, we would like to highlight that our proposed bound is a nice generalization of several existing approaches like IWAE, FIVO, and VRS (see paragraph 3 of related work), i.e., it generalizes several existing VI algorithms in a single framework. In addition to unbiasedness, we also explained how to maximize the bound, the gradient term and its variance reduction, tuning the hyper-parameter $ M(.) $, and discussing the role of $ K $ and $ \gamma $. The above summarizes novel and important contributions of our work, in addition to the alternative unbiased estimation of the marginal likelihood proposed in our paper. Thus, we respectfully disagree that our paper is solving something that is "already tackled," since the above problems are not addressed in either Kudlicka et al. or Schmon et al., and we do not believe these details are straightforward/trivial. Our experimental results confirm that the proposed bound indeed improves VI based bounds like FIVO, IWAE. We again sincerely hope that these aspect will alleviate the concerns from the reviewer.
>
> \textbf{Thus it was not clear to me ... likelihood. }
> We agree that Kudlicka et al. (2020) also presented an unbiased estimator for SMC-PRC. However, there can be multiple ways to construct unbiased estimators for SMC-PRC. We believe our estimator is itself a valuable contribution to the literature in providing users a toolbox of multiple unbiased estimation techniques in SMC-PRC. We stress here again that the main contribution of our work is to obtain an improved VI bound, and the unbiased estimation of SMC-PRC is only one of tools used in the pursuit of this goal.
>
> \textbf{That paper is cited by the authors, but the authors ... "equation (12)" of that paper}
> We respectfully disagree with the reviewer that our work is a trivial modification of "Bernoulli Race Particle Filters" by Schmon, Doucet, Deligiannidis. Please see the clarification below:
>
> (1) "Equation 12 and algorithm 2 are exactly same". In their paper, Schmon et al, implement their "Bernoulli Race" algorithm, for which they have to calculate an upper bound $ c $ for every experiment to bound their intractable weights; there is no closed-form expression. The efficiency of this algorithm is then critically dependent on the tightness of this upper bound, carefully obtained for niche individual problem. In our work, the intractable normalization constant $ Z(.) $ is naturally bounded, and in the event that this bound is too loose, our tuning parameters allow for fast implementation of the dice-enterprise.
>
> (2) In our case, we can control Bernoulli factory's efficiency through a user parameter $ \gamma  $, i.e., our algorithm will take around $ \gamma^{-1} $ (independent of the problem under consideration) steps in general. On the other hand, the approach of Schmon et al. will terminate in around $ \frac{1}{\overline{b} } $ steps (please see page 4 of Schmon et al.) on which they have no control and no guarantees, unlike our method.  Schmon et al. have to not only find a proper $ c $ value (which is known to be challenging outside of toy problems); a loose upper bound can make their method impractical even for 1-dimension. We will request the reviewer to see our discussion on Proposition 1 (page 4) and Section 3.3 explaining the role of $ \gamma $.
>
> (3) Most of Bernoulli factory's existing work is limited to 1-2 dimensions (even  "Bernoulli Race Particle Filters"). However, our approach applies to high dimensional sequences often seen in machine learning. We are not aware of any method that has used the Bernoulli factory for such general settings; therefore, we don't think it is straightforward. To put things in perspective, the maximum sequence length in \textbf{pianomid} data-set is around 36000 with each latent variable $ \{z_{t}\}_{t=1}^{36000} \in \mathbb{R}^{64} $, we are quite sure that approach of Schmon et al. is not applicable here.
>
>
> \textbf{Furthermore, I don't ... of Morina et al}
> Regarding the name "Dice-Enterprise", please see our comments in "Common Response to all reviewers".

---

> > ### Comment · AnonReviewer4 · 2020-11-18
> > **Not convinced at all**
> >
> > I am far from convinced by the authors' reply.
> >
> > First, the comparison to Kudlicka et al (2020) is relevant. Second, there's nothing really novel about using some variant of SMC to perform VI. Entire PhD thesis have been written on the topic, for example that of Christian Naesseth.
> >
> > Third, despite what the authors say their "dice enterprise" in Section 2.2 is in fact exactly the same trick as that presented in the "Bernoulli Race Particle Filters" paper. It's a technique to sample from a Categorical distribution when one has only access to unbiased estimators of the probabilities. It's *exactly* Algorithm 2 in "Bernoulli Race Particle Filters".
> >
> > Proposition 1 of the proposed paper (about the cost) corresponds exactly to an equation appearing in their Section 3.1 "Efficient implementation", that reads
> > \mathbb{E}[C_{j,N}] = \frac{\sum_k c_k}{\sum_k b_k c_k}
> > It does not take much effort to find the bijection between the notation of both papers.

---

> > > ### Author Response · Authors · 2020-11-20
> > > **Summary of Response to AnonReviewer4**
> > >
> > > We sincerely thank the reviewer for engaging in a very stimulating discussion regarding their concerns. While we respect the reviewer's reply to our original response, we also respectfully disagree with their comments regarding Kudlicka et al (2020) and also regarding the "Bernoulli Race Particle Filters" paper. We believe that merely calling our work as a simple application of an SMC technique for VI significantly discredits our contributions. We have provided a detailed response below to the reply by the reviewer, with an earnest hope that our response below will help them see our work in a better light, especially w.r.t. their comments regarding Kudlicka et al (2020) and the ``Bernoulli Race Particle Filters'' paper. We sincerely request the reviewer to go through our response below and we hope the reviewer will consider reassessing their evaluation of our work. We also request all reviewers and the area chair to consider these points in the discussion about our paper.

---

> > > > ### Author Response · Authors · 2020-11-20
> > > > **Detailed Response to AnonReviewer4 Part 1**
> > > >
> > > > \textbf{First, the comparison to Kudlicka et al (2020) is relevant. Second, there's nothing really novel about using some variant of SMC to perform VI.}
> > > >
> > > >  We respectfully disagree with the comment "there is nothing novel about using some variant of SMC with VI." We believe this significantly discredits our work. We are indeed well aware of the Ph.D. thesis of Christian Naesseth. However, if we regard our work as a mere application of SMC to VI (which we believe is not the case), by that analogy, many flexible VI bounds recently proposed in the literature are just variants of sampling-algorithm/SMC and VI. The existing works like Variational Rejection Sampling (VRS) and Variational Contrastive divergence (VCD) could be simplified as just "some variant of sampling with VI." Similarly, papers like FIVO, AE-SMC  could be simplified as only " some variant of SMC to perform VI." In our humble view, this would be a too narrow interpretation of our work. We also believe that coming up with flexible VI bounds requires much more effort than just trying some variant of sampling/SMC to perform VI. The technical nuances of our work are important and should not be disregarded without considering them.
> > > >
> > > >
> > > >  To make things more concrete, we will shed some light on the current work and explain why combining VI with SMC-PRC is a fundamental problem in itself. In the context of VI, learning a flexible family of distribution has always been a significant point of interest, and our work both provides a significantly improved approach, while also generalizing several recently proposed methods in this direction.
> > > >
> > > >  (1) \textbf{Combining VI with sampling}: The existing methods in this area are Variational Rejection Sampling (VRS) and VCD (Variational Contrastive divergence) (for brevity, we are not mentioning other papers). Such algorithms combine VI with a sampling-based algorithm like accept/reject or MCMC to develop a flexible family of distributions.
> > > >
> > > >  (2) \textbf{Using particle approximation in VI}: Such methods leverage importance sampling or SMC (as done in the Ph.D. thesis of Christian Naesseth). The existing papers in this area are FIVO, AE-SMC, VSMC, and  IWAE.
> > > >
> > > >  SMC-PRC, which we consider in our work, is interesting because it essentially combines both these approaches: It uses accept/reject and a particle approximation. Therefore, developing a VI bound for SMC-PRC is interesting because it combines these two areas into a single framework. Consequently, we disagree that SMC-PRC for VI is just some arbitrary variant but a well thought out scheme and is much more interesting. Further, we can express existing work in these two areas as special cases of VSMC-PRC bound.
> > > >
> > > >  (1) N=T=1 our bound reduces to VRS (Variational Rejection Sampling)
> > > >
> > > >  (2) Accepting all particles ($\gamma =1 $) our bound reduces to FIVO, VSMC, AE-SMC
> > > >
> > > >  (3) Using $\gamma =1 $ and avoid resampling, the proposed bound reduces to IWAE.
> > > >
> > > >  (4) N=T=1 and $\gamma =1 $, the bound essentially reduces to standard VB.
> > > >
> > > > Note that the above connections are not made anywhere in the literature, and we believe that shedding light on these connection is also an important contribution of this work. Although we have explained these connections in paragraph 3 of related work, we have highlighted SMC-PRC's importance within VI in the revised version. We would also like to highlight that R1, R2, and R3 have recognized our contribution to the area of VI. Notably, it has been pointed out that the " proposed approach outperforms IWAE and FIVO for a given calculation time" and " The experimental results compare favorably to prior works like FIVO and IWAE" by R1 and R3. Further, the combination of scalable Bernoulli factory with VI has been considered "new and interesting" by R2.
> > > >
> > > >  To summarize the above, there is a well-motivated reason why combining SMC-PRC with VI is desirable and useful. R1, R2, and R3 have considered our work as an essential contribution in the field of VI. Therefore, we strongly disagree that our work is straightforward. We have already mentioned the contribution of Kudlicka et al. (2020) in the revised version. Comparison with Kudlicka et al. (2020) will distract the reader from the main point of our work: to use SMC-PRC to improve VI bound. We will be happy to elaborate upon the positioning of our work w.r.t. Kudlicka et al. (2020) in the final version; an experimental comparison is not possible since the work of Kudlicka et al. (2020) does not perform variational inference unlike our method.

---

> > > > > ### Author Response · Authors · 2020-11-20
> > > > > **Detailed Response to AnonReviewer4 Part 2**
> > > > >
> > > > >  \textbf{ Section 2.2 is in fact exactly the same trick as that presented in the "Bernoulli Race Particle Filters" paper. It's a technique to sample from a Categorical distribution when one has only access to unbiased estimators of the probabilities}
> > > > >
> > > > >  The trick presented in "Bernoulli Race Particle Filters" is a straightforward extension of the two-coin  Bernoulli factory algorithm of Gonçalves
> > > > > et al. (2017a;b), which served as the motivating reference for obtaining our generalized Bernoulli factory.  There is nothing unique about constructing a multinoulli extension of this Bernoulli factory; what we want to stress is that our proposed "trick" is not in the outline of the Bernoulli factory algorithm but in the details that make our implementation considerably faster than the ones in Goncalves et al. and Schmon et.al.
> > > > >
> > > > > Our main trick in the Bernoulli factory implementation that makes our method efficient is the controlling of the acceptance rate $ \gamma $.   The efficiency of Bernoulli factory crucially depends on the intractable coin probability p(i,t) (see equation 8 page 3 of our paper). In contrast to existing approaches of Gonçalves et al., 2017a;b and  Schmon et al., we  control the Bernoulli coin probability $ p^{i}_{t} $ through hyper-parameter $ M $. Using the geometric acceptance rate property of the Bernoulli factory, we can show that the average number of steps taken by our approach is
> > > > >
> > > > >  $   \mathbb{E}[\text{average number of steps}] = \frac{\sum_{i=1}^{N}c^{i}_{t} }{\sum_{i=1}^{N}c^{i}_{t}p^{i}_{t} } \approx \frac{1}{\gamma}. $
> > > > >
> > > > > To learn more about setting $ \gamma $ and $ M $ please see Section 3.3 of the main paper. Therefore, our main "trick" is that we can control our efficiency through a user parameter $ \gamma $. Note that the existing approaches like Gonçalves et al., 2017 and  Schmon et al. have not tried to optimize the Bernoulli factory. Therefore this makes our approach a useful contribution, especially to readers of machine-learning interested in a scalable implementation. Further, for implementing the proposed bound VSMC-PRC on experiments like Toy GSSM and VRNN, we do not have the luxury to use Gonçalves et al., 2017 or  Schmon et al. directly since the dimension of the problem is already very large. In such high-dimensional sequences, faster implementation is a firm requirement.
> > > > >
> > > > >
> > > > > Regarding  proposition 1:
> > > > >
> > > > > Both "Proposition 1" of our paper or "Equation (12)" of Schmon et al. are useful, but straightforward mathematical results. In fact, both these equations could easily be derived given the existing work of the Bernoulli factory. \emph{The geometric nature of Bernoulli factory/multinoulli Bernoulli factory acceptances are well-studied before Schmon et al., for example, Gonçalves et al., 2017, Dughmi et al., 2017, and Flegal et al., 2012}. Therefore, highlighting that  " Proposition 1" is the same as "equation (12)" doesn't make sense, given that it is a well-known result (and we do not claim it as a novelty either). We have also named it a "Proposition" for this precise reason and not as a theorem because it is obvious to see the geometric nature of dice-enterprise.
> > > > >
> > > > >
> > > > >
> > > > >  To make things more concrete, let's consider the notation and  Bernoulli factory from Goncalves et. al.
> > > > > Suppose we have two Bernoulli coins with intractable coin probability $ p_{1} $ and $ p_{2} $ with upper bounding constants $ c_{1} $ and $ c_{2} $. Bernoulli  factory aims to simulate a coin with probability $\frac{c_{1}p_{1} }{c_{1}p_{1}+ c_{2}p_{2} }$.
> > > > >
> > > > >
> > > > > The expected number of steps for simulating a Bernoulli coin is geometrically distributed with the following parameters.
> > > > >
> > > > > \begin{equation}
> > > > >     \mathbb{E}[\text{number of trials} ] = \text{Geom} \left( \frac{c_{1}p_{1} + c_{2}p_{2} } { c_{1} + c_{2} }  \right)
> > > > > \end{equation}
> > > > > It is relatively straightforward/trivial to see that if we use $ K $ coins, the expected time would also be geometrically distributed. For more details please refer to Dughmi et al., 2017.
> > > > >
> > > > >
> > > > > To stress, we repeat again here that our essential contribution with this particular Bernoulli factory is to \emph{scale the Bernoulli factory} for use in higher dimensional problems. Without scalability, it is impossible to develop a VI bound. The similarity is that both Schmon et al. and our work use the Bernoulli factory for resampling, but the efficiency is very different. For clarity, we discuss the contribution of Schmon et al. in Section 2.2 in the revised version, in addition mentioning the related works section.

---

> > > > > > ### Author Response · Authors · 2020-11-20
> > > > > > **Detailed Response to AnonReviewer4 Part 3**
> > > > > >
> > > > > > \textbf{It does not take much effort to find the bijection between the notation of both papers.}
> > > > > >
> > > > > > Our motivating reference for constructing our generalized Bernoulli factory was "Barker’s algorithm for Bayesian inference with intractable likelihoods" by Gonclaves et al. Our notation involving $c$ and $p$ is also borrowed from their work. These are standard notations from Bernoulli factory literature, and using any other notation will unnecessarily confuse the readers. We hope it clears your doubt regarding the "bijection" between our paper and Schmon et al.
> > > > > >
> > > > > >
> > > > > >
> > > > > >
> > > > > > \textbf{It's exactly Algorithm 2 in "Bernoulli Race Particle Filters".}
> > > > > >
> > > > > > Indeed, both Eq (12) and Algorithm 2 are, in principle, straightforward extensions of the two-coin algorithm of Goncalves et al. However, as pointed out before, our "trick" is using $\gamma$ to obtain efficient implementation. Since we aimed to develop a novel, VI bound, unlike Gonçalves et al., 2017 or  Schmon et al.; efficient performance is a strict requirement.
> > > > > >
> > > > > >  Although we have said it before, we will again point out that applying the Bernoulli factory outside of toy example is quite non-trivial. As far as we know, our work seems to be the only paper that has used the Bernoulli factory on general ML models like VRNN.
> > > > > >
> > > > > >
> > > > > > \textbf{Concerns/doubts regarding contribution}
> > > > > >
> > > > > > The aim of the paper was to create a superior VI bound that could perform well as compared to existing works. Although, we have developed an unbiased SMC-PRC estimator, it is still a secondary contribution in contrast to improvement in VI bound and showing its practical viability as our work does. R1, R2, and R3 seem to agree that our proposed estimator performs well as compared to existing VI bounds and is a novel contribution to the field of VI. There is some misunderstanding that our work was a novel contribution to SMC-PRC or the major contribution is only unbiasedness. Although it has been claimed that our work is straightforward extension of Schmon etal, without designing efficient Bernoulli factory, we simply can't construct a VI lower bound. We will request the reviewer to check out our experiments on VRNN to compare and contrast our bound with existing VI algorithms. Even designing a practical algorithm on Bernoulli factory is a non-trivial contribution. This particular aspect of our work is also highlighted by R2. We will sincerely request the reviewer again to note that our proposed methodology is a contribution primarily to the VI literature and not SMC methods. Therefore, the constant emphasis/comparison with SMC approaches like Kudlicka et al. (2020) and Schmon et al is somewhat unjustified. We thank the reviewer for considering our response and again sincerely hope that now they are able to more clearly see and understand the scope and positioning of our work.

---

### Official Review · AnonReviewer2 · 2020-10-26

**Rating:** 7
**Confidence:** 4

**Review:**

Summary:
The submission suggests a new variational bound for sequential latent variable models. Unlike previous work that optimize this bound using ‘standard’ particle filters with unbiased resampling, the new bound is constructed based on a partial rejection control step and uses a dice enterprise for sampling the ancestor variables.

Positives:
The combination of partial rejection control and dice enterprise for variational inference is new and interesting. Particle filters with partial rejection control have been used before for constructing (biased) bounds based on the marginal likelihood. However, using a dice-enterprise step allows for a new unbiased bound which makes it possible to consider a lower bound on the log-likelihood via variational ideas that can be optimized with standard techniques. Empirical experiments suggest that the method outperforms previous work.

Negatives:
Does the complexity of the new bound not scale linearly with K (while K=1 for FIVO)? This seems to be not accounted for in the experiments. Choosing larger N=16 also has a better performance in the FIVO paper.

Recommendation:
I vote for acceptance of the paper. However, I think that the experimental section should be improved.

Comments:
Variational bounds can also be constructed by targeting a smoothing distribution (Lawson et al, 2019) and particle filters with complexity N^2 based on a marginal Fisher identity have been suggested (POYIADJIS et al, 2011) for parameter estimation that avoid estimator variances scaling quadratically in time. I was wondering if there is a connection between such filters and the method suggested here, particularly for K=N?
Can you explain the connection between the variance of the estimator for the normalizing constant obtained from particle filters and the tightness of the variational bound in more details?
Are the signal-to-noise gradient issues for large N or K?
How do the methods in the experiments compare for a larger number of particles?
Is there some useful practical advice on choosing the ratio N/K and gamma?

---

> ### Author Response · Authors · 2020-11-17
> **Response to AnonReviewer2**
>
> Thank you for the helpful comments. Our response to your comments is provided below (for ease of reference, we have highlighted the text from your original comments, while responding to them). Please also note that we have also revised the manuscript incorporating the various comments from the reviewers.
>
> \textbf{Does the complexity of the new bound not scale linearly with K (while K=1 for FIVO)? This seems to be not accounted for in the experiments. Choosing larger N=16 also has a better performance in the FIVO paper. I think that the experimental section should be improved.}
>
> Yes, the time complexity of bound indeed scales linearly with $ K $. However, we wanted to highlight that increasing $ K $ does not significantly improve the bound value despite increasing the time consumption. Therefore, $K>1$ seems unnecessary to us. If we look at the experiments closely, $K=1, \gamma > 0.8$ appears to be a good enough configuration for VI practitioners. We will include this conclusion in the experimental section as well to avoid confusion.
>
> Choosing larger $N$ will improve FIVO but will also enhance VSMC-PRC since it is an SMC bound. We deliberately avoided many particles as it may compromise the gradient quality, as explained by "Tighter Variational Bounds are Not Necessarily Better" by Rainforth et al. and also discussed by AE-SMC (Le et al., 2017).
>
> \textbf{I was wondering if there is a connection between such filters (Lawson et al, 2019, POYIADJIS et al, 2011 ) and the method suggested here, particularly for $K=N$?}
>
> We think that our method is different from the work of Lawson et al, 2019 because we assume that the latent variable $ z_{1:t} $ is conditionally independent of $ x_{t+1:T} $ given $ x_{1:t} $. It would be interesting to consider PRC for such models, though.
>
> \textbf{Can you explain the connection between the variance of the estimator for the normalizing constant obtained from particle filters and the tightness of the variational bound in more details?}
>
> Let the estimator for normalization constant be $Z$. As the variance of $Z$ decreases (through more particles or PRC), we expect $E[\log(Z)]$ (variational bound) to get tighter. However, we would like to highlight that maximizing $E[\log(Z)]$ is not the ideal choice to get a low variance estimator of the normalization constant. The best option is to minimize chi-square divergence "Variational Inference via $\chi$-Upper Bound Minimization" by Dieng et al. Unfortunately, minimizing chi-square divergence is not feasible for large-scale problems.
>
>
>
> \textbf{Are the signal-to-noise gradient issues for large N or K? How do the methods in the experiments compare for a larger number of particles? }
>
> Increasing $ K $ should not cause gradient issues because, even in the limit, the normalization constant $ Z(.) $ still depends quite strongly on variational parameters. Although the same cannot be said for large $N$ because the bound will approach the marginal likelihood reducing the gradient quality. "Tighter Variational Bounds are Not Necessarily Better" Rainforth et al.  explains this phenomenon in far more detail. Although comparing our method with FIVO for a large number of particles would be interesting, it could be problematic in a VI context because it may compromise the gradient quality. We would also like to add that PRC is a greedy strategy; therefore, we can construct some specific scenarios in which FIVO may outperform VSMC-PRC, though we did not find this in practice.
>
> \textbf{Is there some useful practical advice on choosing the ratio N/K and gamma?}
>
> We believe that choosing $K=1$ and $\gamma>0.8$ is a reasonable choice for any $N$. Although we can use large $ K $, the gains are not that much (see our experimental results on VRNN); therefore, $K=1$ seems appropriate. Similarly, choosing low $\gamma$ could be detrimental to the algorithm as it increases the time taken by both the PRC step as well as the resampling step via dice-enterprise. Although we are not sure about this, low $\gamma$ can also reduce the gradient quality as the PRC's improved density may become independent (slightly dependent) of the variational proposal.
>
> We have added this recommendation ($K=1$ and $\gamma>0.8$) in the experiment section to help the readers readily implement our method. Choosing $N$ can be tricky due to the gradient issue, but we believe $ N \leq16 $ (as done in FIVO) should not be that problematic.

---

### Official Review · AnonReviewer1 · 2020-10-27
**Official Blind Review #3**

**Rating:** 6
**Confidence:** 3

**Review:**

### Review update following author discussion

I've read the author responses as well as some of the discussion with the other reviewers. Overall, this is valuable work and I've considered raising my score, but I think a weak accept is appropriate, all things considered.  I've raised the confidence score for my review, as I understand the technical details better now. I think a key strength compared to prior work is the empirical validation of the approach on a variational RNN.  However, the significance of the paper in terms of the novelty of the ideas, both conceptual and technical are overstated in my opinion, hence the weak accept.

One other point of feedback for the final version in case of an accept:

I also share R3's concern about the paper's positioning in relation to the extremely general dice enterprise framework (or, even the bernoulli factory, for that matter) as somewhat misleading for the particular use case in Section 2.2.  The exact same multinomial sampling scheme is in fact more succinctly presented (proposed?) in BRPF [Schmon et. al, 2019] as the "Bernoulli Race" (which the authors have cited).  I would think that the current scheme is a special case of the "Bernoulli race" that uses a particular form of the acceptance probability parameterization similar to prior work (e.g. VRS [Grover et. al. 2018]).

See Section 3, specifically e.g. Eq (10) and Proposition 2 from BRPF [Schmon et. al., 2019], which can be compared to Eq (3) and Proposition 1 in the current submission, respectively.

Minor nit: In step 3 of the algorithm (right below Eq (8)), you use the notation $z_t$ for sampling a new variable from $q_\phi$. However, this $z_t$ has nothing to do with the latent variables that are used in computing the constants $c^i_t$. The way it's written makes it appear as if there's a circular depdendence of the $c_t$ on $z_t$ and then the $z_t$ is again resampled, which changes the $c_t$. For this reason, it maybe better to use a completely unrelated variable for the sample from $q_\phi$ in step 3.


### Key Strengths

The paper puts together several ideas from prior works (partial rejection control/SMC, variational inference, dice-enterprise) and also evaluates these ideas for latent variable sequential state space model benchmarks. The experimental results compare favorably to prior works like FIVO and IWAE and demonstrate that using partial rejection control is beneficial in a variety of benchmarks.

### Key Weaknesses

It seems like a key contribution in terms of novelty/theory is in fixing the bias in prior works using Partial Rejection Control in SMC (e.g. Peters et. al. 2012). More information to support this claim in terms of why the prior estimate is biased would be helpful in assessing the strengths of the paper's contributions (Peters et. al don't seem to explicitly focus on the exact bias for PRC).  Having said that, the experiments seem to show that unbiased gradients are worse than a biased version (Figure 2, left bottom), so it seems like focusing on the exact bias is not that important.


### Additional Comments

* The proof of unbiasedness (Prop 2) says "it is easy to show that Eq (15) is an unbiased estimator" and refers to prior work by [Naesseth et. al] for details. More clarity here would be helpful (especially around the term $Q_{VSMC-PRC}$) for assessing this claim, given that this is one of the main contributions claimed in the paper (besides also commenting on where the bias in prior work is coming from).

* The $Z()$ function first appears in Equation (9), without a prior reference/definition. You might want to introduce this around Eq (4), where the integral appears. There's also a reference to what will later be $Z$ as $p^i_t$ in Eq (8).

* Using partial forms of rejection to proposal distribution samples specifically for variational inference (rather than SMC more generally) has also been considered in prior work [R. Gummadi, "Resampled Belief Networks for Variational Inference", Advances in Approximate Bayesian Inference Workshop, 2014].

---

> ### Author Response · Authors · 2020-11-17
> **Response to AnonReviewer1**
>
> Thank you for the helpful comments. Our response to your comments is provided below (for ease of reference, we have highlighted the text from your original comments, while responding to them). Please also note that we have also revised the manuscript incorporating the various comments from the reviewers.
>
> \textbf{More information to support this claim in terms of why the prior estimate is biased would be helpful in assessing the strengths of the paper's contributions (Peters et. al don't seem to explicitly focus on the exact bias for PRC)}
>
> The ancestor variables will be biased if we use the Monte-Carlo estimate of $\alpha_{t}(.) $ (please see equation 7 of the main paper); this was done in Peters et al. To avoid that, we use a  dice-enterprise to fix the bias issue. Aside from improving the bias issue, the main contribution of our work is developing a superior VI bound  compared to existing bounds like FIVO and IWAE. We can further see the proposed bound VSMC-PRC as a nice generalization of existing approaches like FIVO, IWAE, and VRS  (please see figure 1 and paragraph 3 of related work).
>
> \textbf{Having said that, the experiments seem to show that unbiased gradients are worse than a biased version (Figure 2, left bottom), so it seems like focusing on the exact bias is not that important.}
>
> Yes, we have used biased gradients due to high variance (this problem is not unique to our method; FIVO also suffers from the same issue). However, the bound values presented in the toy example and VRNN are still unbiased. To make things more concrete, let's denote Z to be an unbiased estimator for marginal likelihood p(x); through Jensen's inequality, we can show that E[log Z]<log p(x). The quantity E[log Z] is maximized wrt some parameters. If $ Z $ is biased then we can't say that E[log Z] is a lower bound on marginal likelihood and therefore maximizing it doesn't make much sense. Although unbiasedness is satisfied easily for ELBO and FIVO bound, it is not straightforward to construct unbiased estimators for SMC-PRC. Therefore, fixing the bias issue is essential for combining VI with SMC-PRC.
>
>
>
>
> \textbf{The proof of unbiasednes .... given that this is one of the main contributions claimed in the paper }
>
> Yes, we will include more details about the proof in the supplementary material. We have explained the bias issue above.
>
> \textbf{The  function first appears in Equation (9), without a prior reference/definition}
>
> Thanks for pointing out. We will fix this.
>
> \textbf{Using partial forms of rejection ... }
>
> Thank you for pointing out the work by Gummadi (2014). We have cited it now in the revised manuscript.

---

> > ### Comment · AnonReviewer1 · 2020-11-21
> > **Some additional questions**
> >
> > Thanks for the response. A couple of comments on the response.
> >
> > * "..*The ancestor variables will be biased if we use the Monte-Carlo estimate of  (please see equation 7 of the main paper)*..." <--in this statement, what exactly does "*ancestral variables will be biased*" mean? As you say, the ancestor variables are defined in Eq 7 and it is straightforward to compute their expectation. Can you clarify what is the bias with respect to here? To put it another way, what should the expectation of the RV in Eq (7) have been for it to be unbiased?
> >
> > * I still think the claim in the revised manuscript about novelty from proposing a partial accept/reject mechanism (even for VI) is fully accurate due to the prior works despite the additional citations.

---

> > > ### Author Response · Authors · 2020-11-22
> > > **Response to AnonReviewer1**
> > >
> > > Thanks for your comments. We are a bit unsure about the second point in your comments. Is there a typo in the sentence ".. is fully accurate" where you might have meant to say ".. is NOT fully accurate"? We would appreciate if you could please clarify so that we could respond to this point, and also to your first point accordingly.

---

> > > > ### Comment · AnonReviewer1 · 2020-11-22
> > > > **Yes, sorry for the typo in my comment.**
> > > >
> > > > You are right about the typo in my comment, thanks for checking.
> > > >
> > > > Quoting from the paper:
> > > > "*Although the idea of combining VI with an inbuilt accept-reject mechanism is not new a key distinction of our approach is to incorporate a partial accept-reject mechanism*".
> > > >
> > > > To clarify, my original review comment gave evidence against this claim by pointing out a reference [Gummadi, 2014] to a prior use of partial accept/reject for VI.

---

> > > > > ### Author Response · Authors · 2020-11-23
> > > > > **Response to AnonReviewer1**
> > > > >
> > > > > **"..The ancestor variables will be biased if we use the Monte-Carlo estimate of (please see equation 7 of the main paper)..." <--in this statement, what exactly does "ancestral variables will be biased" mean?**
> > > > >
> > > > > Consider the ideal resampling distribution (equation 7 of the main paper) achieved through
> > > > > dice-enterprise. Let's define $L = \sum_{j=1}^N \alpha_t(z^{j}_{1:t})$ and
> > > > >
> > > > > $ p_{i} =  \alpha_t(z^{i}_{1:t})/L $, we have
> > > > >
> > > > > $ A^{i}_{t} \sim
> > > > > \text{Categorical}(p_1,p_2,...,p_N)
> > > > > $
> > > > >
> > > > > Suppose if we use Monte-Carlo estimate of $ \alpha_{t} $ say $ \widehat{\alpha_{t} }$ than the new ancestor samples (say $ B^{i}_{t} $) would be drawn from
> > > > >
> > > > > $ B^{i}_{t} \sim
> > > > > \text{Categorical}(\widehat{p}_1,\widehat{p}_2,...,\widehat{p}_N),
> > > > > $
> > > > >
> > > > > where $ \widehat{p}_i = \widehat{\alpha}_t(z^{i}_1:t)/\widehat{L} $ and
> > > > >
> > > > > $ \widehat{L}= \sum_{j=1}^N \widehat{\alpha}_t(z^{j}_1:t)$
> > > > >
> > > > > There are various issues with the above equation:
> > > > >
> > > > > (1)  In general it is known that $ \mathbb{E}[X Y^{-1}]\neq  \mathbb{E}[X] \mathbb{E}[Y]^{-1} $, therefore even in expectation
> > > > >
> > > > > $ \mathbb{E}[\widehat{p}_i] \neq p_i $
> > > > >
> > > > > (2) Since we are using $ B^{i}_{t} $ in the place of true ancestor samples we can no longer guarantee the unbiasedness of marginal likelihood.
> > > > >
> > > > > (3)  if $ \widehat{\alpha_{t} }$ has high variance it could lead to improper resampling leading to reduced performance.
> > > > >
> > > > > **I still think the claim in the revised manuscript about novelty from proposing a partial accept/reject mechanism (even for VI) is not fully accurate due to the prior works. Quoting from the paper:"Although the idea of combining VI with an inbuilt accept-reject mechanism is not new a key distinction of our approach is to incorporate a partial accept-reject mechanism".**
> > > > >
> > > > > Thank you for highlighting this particular aspect of Gummadi, 2014. We will discuss the contribution of  Gummadi, 2014 in the revised version. However, we would like to highlight some specific statements from Gummadi, 2014, which differentiates our work.
> > > > >
> > > > > (1) `````````"\emph{``For now, we will only consider jump rules with $ \Delta_{i} = 0 $}'' on page 2 of Gummadi, 2014. We think that the VI approach of Gummadi, 2014 focuses on $ \Delta_{i} = 0 $ whereas our approach is concentrated on $ \Delta_{i} = i-1 $.
> > > > >
> > > > > (2) ``"\emph{An interpretation of the subclass of jump rules for which $ \Delta_{i} $ = 0, is to view them as approximately breaking down a large rejection event into smaller sequential pre-emptive steps, thereby improving the sampling efficiency, possibly exponentially}.'' on page 3 of Gummadi, 2014. Note that whenever the constraint is violated for $ \Delta_{i} = 0 $ we have to start again from $ i = 1 $; This is very similar to "Rejection Control" by Liu et al., 1998. On the other hand, we are using a partial version of rejection control (Peters et al., 2012), which uses $ \Delta_{i} = i-1 $, so we don't restart from the beginning; this particular aspect differentiates our work.
> > > > >
> > > > >
> > > > >  We sincerely thank the reviewer to highlight this important work, it turns out if we use $ \Delta_{i} = i-1  $, $ N =1 $, and $ C_{i}(z_{1:i}) = a(z_{i}|z_{1:i-1},x_{1:t} )$ (equation 2 of our paper), we can recover a particular case of Gummadi, 2014. We will make appropriate changes in the revised version to address your concerns.

---

### Official Review · AnonReviewer3 · 2020-10-29
**SMC for VI**

**Rating:** 7
**Confidence:** 5

**Review:**

This paper describes an SMC algorithm to sample the posterior distribution of latent states $p_\theta(z_{1:T}|x_{1:T})$ in a latent variable models $p_\theta(x_{1:T}, z_{1:T})$. The authors consider a completely general setting (the authors assume Eq.(1) but clearly there is nothing to assume here, this the standard Bayes rule). It is well known that the vanilla SMC sampler is a good candidate for ELBO because it provides an unbiased estimator of the likelihood. But the authors prefer here to use a more sophisticated version of the  SMC algorithm, which features a partial rejection algorithm, which amounts to eliminate proposed particles which are "large enough" likelihood.
It is difficult for an expert in SMC algorithms to understand the algorithm as it is described (one must even guess the meaning of the notations). Equation 3) is rather misleading because it is not understood that one continues the acceptance/rejection sampling step until the sample is accepted [Peters' paper, a little less ambitious in its generality, is much more readable]. Also, the form of the rejection probability does not help to understand the action beyond the scene. The level of generality here is a killer: what the hyperparameter $M(i,t-1)$ means, what is the meaning of this form of rejection probability (we see something like a Barker ratio), this is very puzzling
The rejection probability modifies the mutation kernel and should be taken into account when computing the importance weights (Eqs. (4) and (5)). This implies to estimate a quantity $\alpha_t(z_{1:t}^i)$ which is not tractable. The authors suggest (similar to Peters (2012)) to use a Monte Carlo estimator of these quantities.
To sample the ancestors variables from Eq. (7) with the $\alpha_t(z_{1:t}^i)$ defined in Eq. (6), the authors use the "Dice-Enterprise" algorithm. It does not seem necessary to appeal to such  beautiful algorithm to understand the validity of the algorithm described page 3. Here again, everything is done to frighten the reader and not much is done to explain what is being done...

The results presented are encouraging and shows that the proposed approach outperforms IWAE and FIVO for a given calculation time. This result is clearly interesting and shows that partial rejection helps despite the additional difficulties linked with the intractability which requires an additional layer of complexity.

I like the paper even if I have found it extremely unfriendly to read !

---

> ### Author Response · Authors · 2020-11-17
> **Response to AnonReviewer3**
>
> Thank you for the helpful comments. We apologize if the manuscript was a bit dense at a few places; we will strive to further improve the readability more in the final revision. Our response to your comments is provided below (for ease of reference, we have highlighted the text from your original comments, while responding to them). Please also note that we have also revised the manuscript incorporating the various comments from the reviewers.
>
> \textbf{Equation (3) is rather misleading because it is not understood that one continues the acceptance/rejection sampling step until the sample is accepted [Peters' paper, a little less ambitious in its generality, is much more readable]. Also, the form of the rejection probability does not help to understand the action beyond the scene. }
>
> Thank you for pointing it out; we will fix the mistake. We chose Barker's acceptance probability because we wanted to relate our method with existing work from the literature, i.e., Variational Rejection Sampling (VRS) (as we discussed in the 3rd paragraph of related work). Note that, with Barker's acceptance probability and $ N=T=1$, our method essentially reduces to VRS. Differentiable versions of acceptance probability are desirable because we will optimize the proposed lower bound later on.
>
> \textbf{Authors use the "Dice-Enterprise" algorithm. It does not seem necessary to appeal to such beautiful algorithm to understand the validity of the algorithm described page 3. Here again, everything is done to frighten the reader and not much is done to explain what is being done...}
>
> Regarding the name "Dice-Enterprise," please see our comments in "Common Response to all reviewers." The name "dice-enterprise" is given to any categorical sampling problem, where the success probabilities are intractable. We want to point out that we do not use the exact dice-enterprise mentioned in Morina et. al.; rather we only use the nomenclature established there.
>
> We just wanted to make sure that all relevant work is appropriately discussed. Further, we have tried to analyze the effect of every hyper-parameter ($ K,\gamma$, and $ M $) so that the reader could get more insights into the VSMC-PRC bound. Also, figure 1 of the main paper and 3rd paragraph of related work was added to make the reader more comfortable with the overall algorithm; for example, the connection with VRS, FIVO, and IWAE. We will add more details about the notation so that the article is more readable. We thank you again for your constructive remarks.

---

### Author Response · Authors · 2020-11-17
**Common Response to all reviewers**

We thank all the reviewers for their thorough reading and helpful comments. Please note that we have also uploaded a revision incorporating feedback from the reviewers, which we hope will address the various comments (we will welcome any other suggestions for the final manuscript, should the paper get accepted).

At the outset, we would like to mention that there seemed to be a concern from Reviewer \#4 about the existence of a very recent work by Kudlicka et al. (2020), which we had cited. We would like to strongly emphasize that their work is fundamentally different in various aspects from ours as we highlight below. More importantly, their work is primarily focused on obtaining an unbiased SMC-PRC estimator (and not about VI) whereas our work, as R1, R2, and R3 also have recognized and mentioned in their reviews, is about improving VI through the construction of such an unbiased estimator. In addition, we also demonstrated, through extensive experiments, the effectiveness of such an approach on practical tasks involving VI. We again want to emphasize that Kudlicka et al. (2020) did not consider VI (either in theory or in practice) in the context of SMC-PRC and therefore the scope and positioning of our work is significantly different from theirs. We have also clarified this point in the revised manuscript. We sincerely hope that it will help address the concerns, especially by R4.

Below is our response to some of the main questions for all reviewers, also highlighting again some of they key aspects of our work. In addition, we separately also provide individual responses to each reviewer's questions/comments.

(1) The primary goal of our work is to provide a superior VI bound (VSMC-PRC) that performs better than existing bounds like FIVO and IWAE. In addition to this, our work is a nice generalization of several existing bounds like FIVO, IWAE, and VRS (Variational Rejection Sampling).

(2) To develop a VI bound, we constructed a novel SMC-PRC estimator, which is unbiased, a contribution that is important in itself. Although, recently, Kudlicka et al. (2020) also provide a solution to this same problem, our proposed estimator is quite different from theirs and, therefore, a significant contribution. Note that Kudlicka et al. samples one extra particle and keeps track of the number of steps used in PRC for every timestep to construct an unbiased estimator. The particle's weight mentioned in  Kudlicka et al. are tractable as they don't consider the effect of normalization constant $ Z(.) $ (see equation 3 of the main paper). On the other hand, we have taken $ Z(.) $ into consideration, making the particle's weight intractable. Since the weights don't have a closed form expression, sampling of the ancestor variable becomes non-trivial. To address this, we use the dice-enterprise.

(3) It has been pointed out by R4 that we should compare our work with Kudlicka et al. (2020) which however is not a VI technique, and therefore a comparison would not be appropriate. Again, please note that our goal is to improve VI through SMC-PRC (leading to our approach VSMC-PRC) and not improve SMC-PRC itself. Although it is an interesting premise for future work, comparing VSMC-PRC with Kudlicka et al. (2020) will distract from the main point of our paper, since the latter is focused on improving the VI bound. We sincerely request the reviewers (especially R4 who has this concern primarily) to evaluate our method as providing an improved VI bound instead of considering our work as an improvement in SMC-PRC itself.

(4) Bernoulli factories, prior to our work, has always been used in one-dimensional toy examples/diffusion models due to scalability and efficiency issues.  We believe our work will draw attention to this simple yet powerful, elegant tool  and  help other ML researchers to solve such intractable problems arising in VI. To the best of our knowledge, there is no prior work that has used Bernoulli factories for such a general case like VRNN; therefore, we believe this aspect to be a significant contribution as well.


There are some concerns about why we have named our method dice-enterprise instead of the Bernoulli factory (R3, R4). A Bernoulli factory is defined as an algorithm that, given a function $f:(0,1) \to (0,1)$, produces an $f(p)$-coin using $p$-coins. Since the outcome set is binary, the nomenclature \textit{Bernoulli} factory is appropriate. However, this terminology is inappropriate when the outcome set is larger than 2.

With this motivation, Morina et. al. (2019), introduced the nomenlcature of a dice-interprise.
In Equation (1) of Morina et. al. (2019), they define the dice-enterprise as an algorithm that, given a function
\[
f:\Delta^m \to \Delta^v\,,
\]
for $m,v \geq 1$, samples $f(p)$-dice with a $p$-dice. Our algorithm in Section 2.2 is a particular instance of a dice-enterprise with $m = v =  N-1$. Our dice-enterprise will be a Bernoulli factory only if the number of particles is 2.

---

### Decision · Program_Chairs · 2021-01-07
**Final Decision**

**Decision:**

Reject

**Comment:**

This paper explores the use of partial rejection control (PRC) for improved SMC-based variational bounds. While an unbiased SMC variant with PRC has been previously introduced by Kudlicka et al. (2020), this work introduces innovations that can help apply such ideas to variational inference. These bounds result in improvements in empirical performance.

This paper was heavily discussed, with significant engagement by both the authors and the reviewers. Most reviewers recommended acceptance of this paper, with one reviewer (R4) recommending against acceptance. R4's central concerns regard the novelty of the proposed approach and its positioning relative to the existing SMC literature. The authors argued vigorously in the comments that this paper should be judged as a contribution to the VI literature and not the SMC literature.  Unfortunately, I will recommend that this paper is rejected. It is my opinion that R4's concerns were not fully addressed.

On the one hand, I agree with the authors that there is significant value to be had in exploring variants of SMC for VI. Indeed, some prior art, like FIVO and IWAE, contributed little to the Monte Carlo literature. I believe that these were good contributions.

On the other hand, I am concerned that the current draft does not clearly circumscribe its contributions. I read the sections that disuss the works of Schmon et al. (2019) and Kudlicka et al. (2020), and the writing did not leave me with a clear enough sense of the differences. I also read the abstract and introduction of the paper. The introduction of the paper positions this work clearly within the VI literature, but does not clearly discuss prior SMC art, e.g., it does not cite Kudlicka et al. (2020). Despite citing rejection control for SMC, the writing of the abstract and introduction left me with the impression that this work was the first to introduce *unbiased, partial* rejection control for SMC. I believe that impressions matter and that the machine learning community should be generous to adjacent communities when assigning credit.

I realize that my decision is a matter of taste. I also want to say that I am confident that the authors have a clear sense of where their contribution sits, and I suspect that it is a valuable contribution. However, I cannot recommend the draft in its current form. If this is a contribution to the VI literature, as the authors argue, then the authors should not hesitate to give full credit to prior SMC art. My reading of the current draft still leaves me confused about which aspects of the SMC estimator are actual contributions.